# Auditory and sexual preferences for a father's song can co-emerge in female Bengalese finches

**Tomoko G. Fujii**[1], **Kazuo Okanoya**[1,2]*

**1** Department of Life Sciences, Graduate School of Arts and Sciences, The University of Tokyo, Meguro-ku, Tokyo, Japan, **2** Behavior and Cognition Joint Research Laboratory, RIKEN Center for Brain Science, Wako, Saitama, Japan

\* cokanoya@mail.ecc.u-tokyo.ac.jp

**Data Availability Statement:** All relevant data are within the following repositories: DOI: 10.6084/m9. figshare.14677650 (URL: https://doi.org/10.6084/ m9.figshare.14677650) and DOI: 10.6084/m9. figshare.14677572 (URL: https://doi.org/10.6084/ m9.figshare.14677572).

## Abstract

Birdsong is an important communication signal used in mate choice. In some songbird species, only the males produce songs. While the females of those species do not sing, they are sensitive to inter- and intra-species song variations, and the song preferences of females depend on their developmental experiences and/or genetic predispositions. For example, in Bengalese finches and zebra finches, adult females prefer the song to which they were exposed early in life, such as the father's song. In the current study, we aimed to test whether the preference for the father's song, as reported in previous Bengalese finch studies, can be interpreted as a mating preference. For this purpose, the subjects were raised exclusively with their family until they became sexually mature and then tested as adults. We measured copulation solicitation displays during playbacks of the father's song vs. unfamiliar conspecific songs and found that across individuals, the father's song elicited more displays than other songs. In addition, we analyzed if a bird's response to a given song could be predicted by the level of similarity of that song to the father's song. Although the birds expressed more displays to songs with greater similarity to the father's song, the effect was not statistically significant. These results suggest that female Bengalese finches can develop a strong mating preference for the father's song if they are exclusively exposed to the father's song early in life. However, it is not clear if such a preference generalizes to other cases in which birds are exposed to multiple male songs during development. In order to fully elucidate the possible contribution of experience and genetic factors in the development of female song preference in this species, future studies will need more detailed manipulation and control of the rearing conditions, including cross-fostering.

## Introduction

Song is an important communication signal used for mate choice in songbirds [1]. In some species of songbirds, only males produce songs. While females do not sing, they are sensitive to inter- and intra-species song variations and may change their behavior depending on features of the song that reflect the sex, species, and condition of the signaler [2]. For example, as is generally the case for acoustic courtship signals across taxa, each songbird species has its

**Funding:** This work was supported by MEXT/JSPS KAKENHI Grant (https://www.jsps.go.jp/index. html), Numbers 17J07023 & 21K20282 to TGF, and 17H06380 & 20H00105 to KO. The funders had no role in study design, data collection and analysis, decision to publish, or preparation of the manuscript.

**Competing interests:** The authors have declared that no competing interests exist.

own song characteristics, and female birds selectively respond to conspecific songs over heterospecific ones [3–5]. In the case of zebra finches (*Taeniopygia guttata*), this sensitivity to species-specific song features seems to be a common tendency across individuals that develops independently of auditory experience [6, 7]. However, developmental experience plays a significant role in shaping other aspects of female song preference. A cross-fostering study in the zebra finch demonstrated that recognition of a subspecies based on song depends on auditory experience with the foster father's song [8]. Early-life song exposure in this species is also critical to be able to discriminate song quality [7] or song performance related to social context [9].

In addition, developmental experience might be an important source of individual differences in song preference. Previous studies in zebra finches and Bengalese finches (*Lonchura striata* var. *domestica*) found that females prefer songs to which they were exposed early in life, including the father's song [10–15]. These studies clearly show that female birds acquire and retain an ability to discriminate their (foster) father's song from other songs, but it is not clear whether such a preference for the father's song is sexually motivated or not. On one hand, it may be reasonable to interpret this preference as an expression of sexual imprinting as previously discussed in some literature [10, 14, 16]. On the other hand, however, we cannot exclude the possibility that such a preference is a general selectivity to a familiar stimulus rather than a mating preference. In these previous studies, preference for the father's song was measured by behaviors such as approach to the sound source [10, 11, 15] or operant behaviors associated with song playback [13, 14], which are not necessarily relevant to a reproductive context. Although there are some studies in song sparrows or zebra finches showing that song preference measured by operant conditioning is correlated with song preference measured by sexual displays [17] and preference for the male singer [18], or that the time spent with a male is a good indicator of mate choice [19], it is still open to question if such correlation generally applies to preference for the father's song in other species. Likewise, it is not self-evident in all species that preferring a trait similar to that of the father even results in a female's reproductive success [20–22]. Instead, it could lead to infertility of offspring if birds inbreed with close relatives due to such a preference. Some studies in songbirds and other bird taxa indicate that memorizing the traits of parents is a possible strategy for kin recognition to avoid inbreeding [23, 24]. Therefore, it is important to elucidate whether this preference for the father's song as reported in previous studies can be regarded as a mating preference. Doing so will allow us to interpret the individual differences in song preference that have been observed in laboratory studies, and will eventually provide a deeper understanding of female mate choice.

In the current study, we aimed to test whether adult female Bengalese finches show sexual responses to their father's song when they are reared with their family members until they reach sexual maturity. In previous studies in this species, females were kept with their parents and siblings until about 120 days post hatch and later tested if they prefer their father's song over unfamiliar conspecific songs by operant conditioning or phonotaxis [14, 15]. To meet the research purpose, here we reared our subjects in the same social conditions as these studies, and measured copulation solicitation displays (CSDs) in response to song playback. CSD is a typical posture that females perform when they accept male courtship, and researchers have utilized it as a reliable index of sexual motivation in response to songs [17, 25, 26]. We also analyzed if song preference in our subjects could be predicted by the similarity of the stimulus to the father's song.

## Methods

### Animals

We used 10 adult female Bengalese finches as subjects for the song preference test. They were obtained from 8 different clutches in our laboratory colony. For breeding, 15 adult finches

were used (7 males and 8 females; one male was paired with 2 different females successively). The 15 adults were either bred in our laboratory or purchased from a commercial breeder. The female subjects were raised by both parents and housed with their families (parents and siblings) in a home cage (size: 30 cm wide × 24 cm deep × 33 cm high) until approximately 120 days post hatch (dph). This age was chosen based on similar song preference studies previously published by our laboratory [14, 15]. Each home cage was placed in a colony room but visually isolated from one another. The female subjects were kept in a single-sex group cage (37 × 42 × 44 cm) after separation from the parents and male siblings (if any). The number of birds kept in a group cage ranged from 6 to 14. All subjects were sexually mature at the beginning of the experiment (mean age = 279 dph, range = 164–376 dph). The females were kept in the group cage until the beginning of the experiment. During this time, they could hear and see other conspecific males kept in other cages in the same room, but they could not physically interact with them. The subjects' fathers were housed in a different room so that the females no longer had acoustical or social access to their fathers after 120 dph. No birds had experienced breeding prior to the current study. For the preparation of song stimuli, 19 adult (> 180 dph) male Bengalese finches were used, 7 of which were fathers of the subjects described above. Of these males, 7 birds (including 4 fathers) were bred in our laboratory. None of these males were siblings, and they all had distinct tutors. The remaining 12 birds (including 3 fathers) were purchased from a pet supplier, so there is no way of knowing their tutoring history. However, judging from the song recording data, the songs of these 12 birds were dissimilar enough to assume that they did not learn their songs from the same tutor.

All birds were kept under a 14:10 h light:dark cycle with food and water provided ad libitum. They were also fed oyster shell and greens once a week. The temperature and humidity were maintained at around 25°C and 60%, respectively. After finishing all the experiments, the subjects, their parents, and the birds used for stimulus recordings continued to be housed in our colony for additional research. All experimental procedures in this study were approved by the Institutional Animal Care and Use Committee at the University of Tokyo (permission #27–9 and #2020–2).

## Song preference test

Briefly, we designed a playback experiment to test if females perform more CSDs to their father's song compared to unfamiliar conspecific songs. The overall schedule of the experiment is as follows: the birds first received a hormone implant, then they were moved to the experimental environment for acclimation and isolation from males, and finally they were tested with song playback (Fig 1(A)).

**Hormone implantation surgery.** Female songbirds seldom perform CSDs in response to song playback alone in the laboratory, but studies have shown that subcutaneous implantation of estradiol increases this behavior [25]. Thus, we adopted this hormone implantation method in our experiment and confirmed that the method did indeed increase CSDs in response to song playback (S1 File). Following a previous study using the same bird species [26], we administered 17β-estradiol (E2758, Sigma Aldrich) in silastic tubing with a 1.0 mm inner diameter (100-0N, Kaneka medical products) containing 8 mm of hormone in powder form. During the surgery, a bird was manually restrained, and lidocaine (Xylocaine, AstraZeneca) was applied onto the skin for local anaesthesia. A small incision was made in the skin on the bird's back and a tube filled with hormone was inserted subcutaneously. The test period started 6–10 days after the surgery (Fig 1(A)). After finishing all the tests, local anaesthesia was again applied to the bird's back and the hormone tube was excised. From the day of implantation, we continued to monitor the physical condition of the birds every day until a few days after the tube removal.

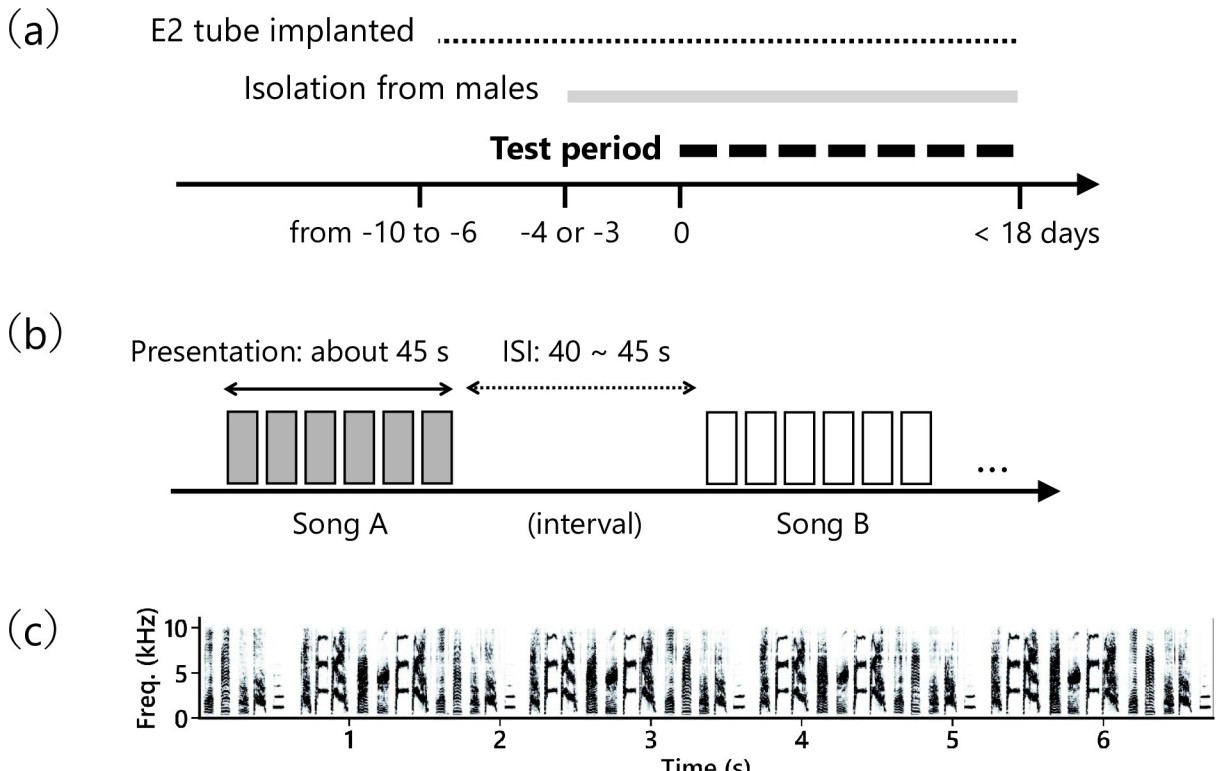

**Fig 1. Experimental schedule and stimuli.** **(a)** Overall schedule of the experiment. Each line above the horizontal axis indicates the period in which birds were under that manipulation. The start of the song playback test was set as day zero. **(b)** A schematic diagram of the stimulus presentation schedule in a single test. Five songs, each composed of 6 renditions (6 rectangles in the figure) were played. **(c)** A spectrogram of one rendition of the song stimulus is shown as an example.

**Preparation and presentation of song stimuli.** In the song preference tests, we presented 5 different conspecific songs to each subject: the father's song and 4 unfamiliar songs. These stimuli were chosen from a pool of songs recorded from 19 males (see below). Females had never been exposed to these 4 unfamiliar songs prior to the experiment. We presented the song of each subject's father to 1 to 4 other subjects (that hatched in a different clutch) to exclude the possibility that a particular characteristic of a given song generally attracts females, independent of the father-daughter relationship. The details of the stimulus sets are summarized in the supporting information document (S2 File).

To prepare stimuli, we recorded songs from 19 adult (> 180 dph) male Bengalese finches in a soundproof chamber through a microphone (PRO35, Audio-technica) fixed at the top of the cage. For the 7 birds that were the fatherss of the subject females, song recordings were conducted before these birds were used for breeding. The signal from the microphone was amplified by a preamplifier (QuadMic, RME) and digitized by an audio interface (Delta66, M-AUDIO) at 16 bits with a 44.1 kHz sampling rate and saved on a computer. All recordings were undirected songs, as each bird was kept alone in the chamber during recording. For each male singer, we randomly selected 6 renditions of songs that were recorded without any background noise (Fig 1(C)). The duration of a single song rendition was 7.04 ± 0.64 (mean ± *s.d.*) seconds (range: 5.41–8.55; sample size: 6 renditions × 19 singers = 114), and the total duration of 6 song renditions was 42.24 ± 0.29 (mean ± *s.d.*) seconds (range: 41.78–42.83; sample size: 19). The sound waveform was first band-pass filtered at 0.5–10 kHz. We normalized the sound amplitude by the standard deviation of the waveform. This normalization was done within a

single song (i.e., between the 6 renditions) as well as between different songs. For each singer, a song was composed of 6 song renditions presented with a 700-ms interval between each rendition (thus approximately 45 seconds in total, Fig 1(B)). The order of the 6 renditions within a song was randomized each time. Song stimuli were played back from a loudspeaker (MM-SPL2N2, SANWA Supply) positioned next to the test cage. The equivalent continuous A-weighted sound pressure level was measured at a point in the test cage that was 22.5 cm away from the speaker and adjusted to approximately 67 dB using a sound level meter (NL-27, Rion) with 'fast' time weighting.

**Environment and procedure for song preference tests.** Preference tests were conducted in a test cage (30 × 24 × 33 cm) placed in a sound attenuation room (163 × 163 × 215 cm). The light:dark cycle, temperature, and humidity of the testing environment was the same as in the colony room. Subjects were moved to the test cage in a group of either 3 or 4 birds (10 birds were divided into 3 groups) at least 3 days prior to the first test. The purpose of this was to acclimate the birds to the testing environment as well as to isolate them from the males both socially and acoustically. The cage mates stayed together in the test cage during the majority of the testing period. However, just before a preference test, birds other than the test subject were caught and moved to another soundproof chamber until the test was completed. When moving them back to the test cage, we minimized handling to reduce stress to the birds. As we tested 1 bird or 2 birds in the morning of a day and each bird was tested once every other day, each bird was handled once or twice per day during the test period. A subject was isolated in the test cage 30–60 minutes before the start of song playback until the end of the test. All cage mates were returned to the test cage as soon as the test was finished.

In a single test, 5 songs (each consisting of 6 renditions) were played back in a random order (Fig 1(B)). The inter-stimulus-interval was a random value ranging from 40 to 45 seconds. Thus, the duration of one test was approximately 450 seconds. Since some birds did not perform any CSDs in a test, the following criterion was set. We defined a test as "responsive" if the subject expressed CSD(s) to at least one of the five stimuli. We continued testing each female until 5 such responsive tests were obtained for that individual. One out of the 10 birds failed to meet the criterion due to health issues. We ceased testing that bird after 4 responsive tests. The total number of tests required to reach the criterion varied between individuals (mean = 6.3 tests range = 5–9 tests, see S3 File for the individual data). Each bird was tested once every 2 days to avoid habituation to stimuli. Thus, the test period for a subject lasted for 10–18 days, depending on the number of tests required to reach the criterion. All the tests were conducted in the morning (7:00AM– 12:00PM).

**Recording and quantification of behavior.** A web camera (BSW200MBK, Buffalo) recorded the movement and posture of subjects during the tests. The frame rate and sampling rate for video and audio recordings were 16 Hz and 44.1 kHz, respectively. The experimenter analysed the videos with the sound muted and had no access to any information regarding stimulus presentation order until the end of analysis, in order to minimize the effect of observer bias in behavioral quantification. For each stimulus in each test, we recorded CSDs performed during the song presentation period. A typical CSD could be recognized relatively easily: the bird usually titled her head and chest down, with her tail feather raised and quivering [27], as previously described in other literature [26, 28]. We defined a posture as CSD if it contained all these features and excluded any ambiguous movement that lacked any one component (e.g., tilted head, chest down, tail feather raised but not quivering). In two previous studies where CSDs were measured in female Bengalese finches, the authors quantified the response by counting the frequency of CSD bouts [3, 26]. The study briefly described that the typical duration of a CSD was 2 seconds, but did not use this duration nor any other measures of response intensity in the analysis [26]. Another study in the Bengalese finch reported

whether or not a bird expressed CSD(s) to each stimulus [29]. In our experiment, some birds continually expressed several bouts of CSDs with individual variations in duration. However, 3 birds out of 10 subjects did not show displays to any of the unfamiliar songs. Thus, we analyzed responses using 2 indices: 1) whether or not a bird performed CSDs in a trial (a song presentation in a test) and 2) the total duration of CSDs observed during stimulus presentation. In the former analysis, data from all tests (regardless of 'responsive' or not) in all subjects was used. The response rate for each stimulus was calculated as the number of tests in which a bird showed CSDs to that song divided by the total number of tests to compare the overall response frequency between stimuli. We used this rate for description and figure illustration purposes but used the actual number of tests showing CSDs in the GLMM analysis detailed below. For the analysis of duration, data from 7 birds that expressed CSDs to both the father's song and unfamiliar songs was used, because 3 out of 10 birds did not show CSDs to any of the 4 unfamiliar song stimuli. To evaluate the response intensity within a trial, we calculated the total duration of CSD(s) observed in a single trial for each stimulus. Each trial consisted of about 45-seconds presentation of a given song stimulus. This total CSD duration per trial for each stimulus was averaged across all trials throughout the test series. Rather than including trials in which no CSDs were shown as a duration of 0 seconds, we excluded these trials from the calculation of mean total duration. For 2 birds that showed CSDs to more than 1 unfamiliar song, mean total duration was calculated by pooling all trials of the different unfamiliar songs.

## Evaluation of song similarity

If specific acoustical features of the father's song stimulate the females' auditory system to induce CSDs, it can be expected that a bird's response to a given song is predicted by the similarity between that song and the father's song. To test this possibility, for each unfamiliar song stimulus, we calculated the acoustical and temporal similarity to the father's song and analysed if these similarity indices predicted CSD frequency. For the evaluation of acoustical similarity, we applied the similarity measurement function (with asymmetric and time-courses mode) in Sound Analysis Pro 2011 [30] to the wave file data of song stimuli used in our song preference tests. For the similarity calculation, each song rendition was divided into 3 segments (duration of a segment = $2.00 \pm 0.11$ (mean $\pm$ *s.d*). seconds). Thus, the total number of files per stimulus was 18 (3 segments $\times$ 6 renditions). For every combination of segments between an unfamiliar song and the father's song ($18 \times 18 = 324$ combinations), the similarity value was computed and the mean value across these combinations was used as a representative value. To evaluate temporal similarity, we measured the song tempo, which is defined as the number of syllables divided by the duration from the first syllable onset to the last syllable offset. The mean song tempo of 6 renditions was used for each song. We then calculated the tempo difference (absolute value) between a given unfamiliar song and the father's song.

## Statistical analyses

To statistically test if females showed more CSDs to the father's song compared to unfamiliar songs, we fitted a generalized linear mixed model (GLMM) to a dataset [31] of all tests from all subjects. In the model, the dependent variable was the occurrence of CSDs in a given trial. The bird's response was binarily coded as 0/1 (not performed/performed), respectively. The type of song stimulus (unfamiliar/father's coded as 0/1) and song presentation order within a test (1–5, a numerical variable) were the fixed effects of independent variables. Subject ID (10 levels), stimulus ID (19 levels), and test numbers (9 levels) were also included as random effects. The binomial distribution with a logit link function was specified to represent the probability

distribution. In addition, Wilcoxon's signed-rank test was used to compare the CSD duration between the father's song and unfamiliar songs ($n = 7$) with significance level $\alpha = 0.05$.

We next analyzed the possible effect of song similarity in the prediction of female response using a GLMM. In this analysis, we used trials of unfamiliar song playbacks from 7 birds that responded to at least 1 unfamiliar song [31]. The total number of trials was 176. The response rate to unfamiliar songs was low, and birds only responded in 17 trials. We fitted a model to test if acoustical and/or temporal similarity to the father's song predicts female response to unfamiliar songs. The dependent variable was the occurrence of CSDs coded as 0/1 (not performed/performed), and binomial distribution with logit link function was specified. The SAP % similarity value, absolute value of the tempo difference, and song presentation order were the fixed effects of independent variables. Subject ID (7 levels), stimulus ID (17 levels), and test numbers (8 levels) were included as random effects.

GLMM estimation and statistical tests were conducted using R version 4.0.2 [32]. We used lme4 [33] and lmerTest [34] packages for GLMM analysis. In all the GLMM analysis, models were fitted using maximum likelihood estimation (Laplace approximation). For estimating the coefficients and their standard errors, $z$-value (Wald statistics) and $p$-value were calculated and reported in the results. For plotting data and fitting logistic regression curves, a Python-based package (SciPy version 0.19.0) was used.

## Results

We first examined if female Bengalese finches performed more CSDs in response to playback of the father's song compared to unfamiliar conspecific songs. In all of the subjects we tested, the father's song elicited more frequent responses compared to the 4 other songs. The mean response rate to the father's song and the second-preferred unfamiliar song was $0.790 \pm 0.209$ and $0.223 \pm 0.193$ (mean $\pm$ s.d.), respectively (Fig 2(A)). Although directly comparing different studies must be done with caution, the response rate of our estradiol-implanted subjects is consistent with the results from another study with a similar design [26]; here, the authors

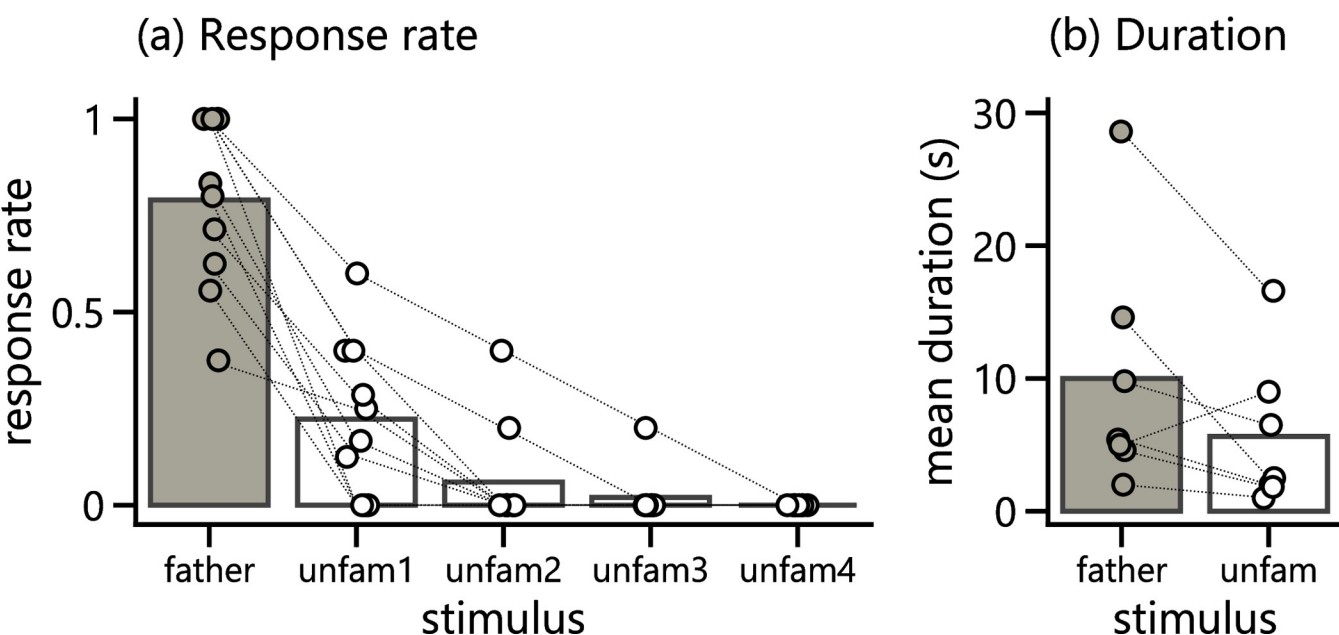

**Fig 2. Females performed more CSDs to their father's song. (a)** Response rate of CSDs to each stimulus ($n = 10$). Five songs are aligned in descending order of response rate. **(b)** Mean total duration of CSDs expressed during presentation of the father's song and unfamiliar songs ($n = 7$). In both panels, dots connected with broken lines show individual data, while bars indicate the population mean.

**Table 1. Effect of song type on females' response.**

| Independent variables | Estimates | | z-value | p-value |
|---|---|---|---|---|
| Fixed effects | coefficient | s.e. | | |
| Intercept | -1.957 | 0.598 | | |
| **song type (unfam/ father = 0/1)** | **4.499** | **0.544** | **8.278** | **< 0.001** |
| **presentation order (1–5)** | **-0.346** | **0.151** | **-2.297** | **0.022** |
| Random effects | variance | s.d. | | |
| subject ID (10 levels) | 1.002 | 1.001 | | |
| stimulus ID (19 levels) | 0.000 | 0.000 | | |
| test number (9 levels) | 0.234 | 0.484 | | |

Estimated parameters of the GLMM are shown. Independent variables with a p-value less than 0.05 are indicated in boldface.

presented 8 unfamiliar conspecific songs (3 times each) to female Bengalese finches and reported that the birds showed on average 2 bouts of CSDs to their most preferred song and no CSDs to less preferred songs. In our study, we statistically analyzed the differences in response frequency between stimuli using a GLMM. The estimated parameters indicated that the type of song stimulus (unfamiliar/father's) was the strongest predictor of the birds' response ($\beta \pm s.e.$ = 4.499 ± 0.544, $z$ = 8.278, $p < 0.001$; Table 1). The presentation order within a test also significantly affected the birds' response, although the estimated effect was relatively smaller than that of the song type ($\beta \pm s.e.$ = -0.346 ± 0.151, $z$ = -2.297, $p$ = 0.022). We also compared the total duration of CSDs per trial between stimuli. The mean total duration of CSDs during song presentation was longer for the father's song than for unfamiliar songs in 6 out of 7 birds that responded to both the father's song and unfamiliar songs. Although the population mean indicated a tendency for longer duration CSDs to the father's song (mean ± $s.d.$ father's: 10.00 ± 8.51; unfamiliar: 5.63 ± 5.23; Fig 2(B)), the difference in CSD duration between song types was not statistically significant ($W$ = 3, $p$ = 0.156).

We next conducted a post hoc analysis on whether similarity of a song stimulus to the father's song predicted responses in the 7 birds that expressed CSDs to both song types (Fig 3). Among the independent variables included in the GLMM, presentation order within a test was the strongest predictor of the response (Table 2). As with the result in Table 1, a song was more likely to elicit CSDs when it was presented earlier in a test (Fig 3(C)). Although there was a tendency for unfamiliar songs to elicit more frequent CSDs when these songs had greater similarity to the father's song (higher SAP % similarity, lower tempo difference, Fig 3(A) and 3 (B)), this result was not statistically significant (Table 2). The overall results imply that female Bengalese finches perceive their father's song as an attractive courtship signal if it is the only tutor song, but we cannot conclude whether their mate choice is affected by song features that are shared with their father's song.

## Discussion

Previous studies in Bengalese finches and zebra finches have found that female birds develop a long-lasting preference for the song(s) they hear early in life, including their father's song [10–

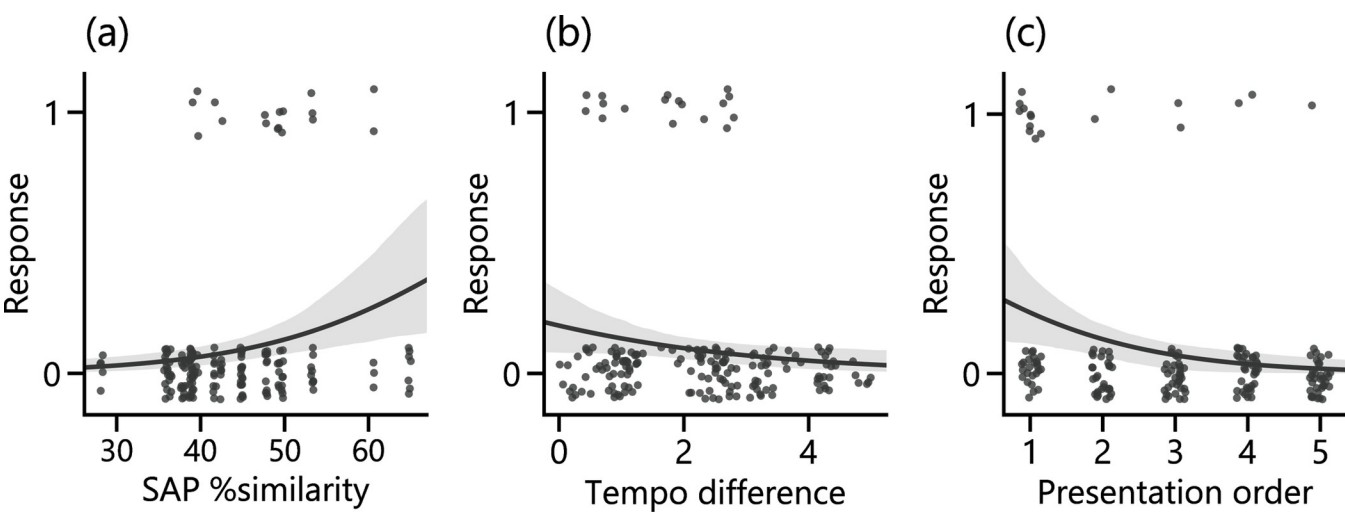

**Fig 3. Analysis of song similarity.** CSD response was plotted against the **(a)** acoustical similarity (SAP % similarity), **(b)** temporal similarity (absolute value of tempo difference) of a given unfamiliar song to the father's song, or **(c)** presentation order within a test (1–5). In all the panels, 0 and 1 on the vertical axis means 'not performed' or 'performed', respectively. Each data point corresponds to one trial. Thus, there are 176 data points from 7 birds in total (17 points are plotted on 'performed'). The lines and surrounding bands are the logistic regression curve with 95% confidential interval computed by bootstrapping method (iteration $n$ = 1000).

15]. These studies of female preference specifically for the (foster) father's song measured general behaviors such as phonotaxis or operant responses, which are not necessarily relevant to a mating context. In several other species, studies have shown similar preferences when measuring both general behaviors such as phonotaxis or operant responses and more mating-relevant behaviors such as CSDs or approach to live males [17–19]. Thus, it seems likely that preference specifically for the father's song may also be in the context of mating. From a functional viewpoint, however, it is not clear if such a preference is an expression of sexual motivation in females. If females use male songs for inbreeding avoidance, it is predicted that they should rather disfavor song features that are very similar to the father's song. In the current study, we tested whether they are sexually attracted to their father's song in case it was the only song tutor. To do so, we administered estradiol to measure CSDs in response to song playbacks. Female Bengalese finches were reared exclusively with their family until about 120 dph and then recruited for song preference tests. We found that birds showed greater frequency of CSDs to the father's song than to other unfamiliar conspecific songs (Fig 2(A)). In addition,

**Table 2. Effect of song similarity on females' response to unfamiliar songs.**

| Independent variables | Estimates | | *z*-value | *p*-value |
|---|---|---|---|---|
| Fixed effects | coefficient | s.e. | | |
| Intercept | -3.369 | 2.195 | | |
| SAP % similarity | 0.068 | 0.039 | 1.745 | 0.081 |
| tempo difference | -0.196 | 0.293 | -0.667 | 0.504 |
| **presentation order (1–5)** | **-0.684** | **0.242** | **-2.828** | **0.005** |
| Random effects | variance | s.d. | | |
| subject ID (7 levels) | 0.291 | 0.540 | | |
| stimulus ID (17 levels) | 0.000 | 0.001 | | |
| test number (8 levels) | 0.288 | 0.536 | | |

Estimated parameters of the GLMM are shown. Independent variables with a *p*-value less than 0.05 are indicated in boldface.

the total duration of CSDs was longer for the father's song compared to unfamiliar songs in the majority of birds (Fig 2(B)). However, the difference in duration between song types was not statistically significant, and it is still a question as to whether female birds exhibit differences in display intensity or degree depending on the song stimuli. In addition, estradiol treatment may interact with acoustical experiences to modify the song preference in zebra finches [35]. Yet, as the selectivity to the father's song is consistent with the results of phonotaxis or call back assay without hormonal treatment [14, 15], it is less likely that the current result is an artefact of the presence of estradiol. Taken together, the overall results suggest that the subject females were sexually motivated when exposed to the father's song. It must be noted here that such a highly selective response might be captured in this unique experimental setting, but not necessarily in other social environment as discussed later in this section.

The song preference observed in our study might be regarded as an expression of sexual imprinting on the father's song. This possibility has already been referred to in an earlier study of captive zebra finches [10] and was demonstrated in the case of song preference at a (sub) species level [36]. Although females preferred songs of the same species as their foster father, there has been no conclusive experimental demonstration showing a sexual preference for the specific song (i.e., selective expression of *sexual displays* to the very song they heard during development) that females were exposed to during rearing. Likewise, field research in Darwin's finches (*Geospiza*) reported female mate preference consistent with sexual imprinting on the father's song [16]. However, the effect of experience with the father's song on later mate preference, if any, may depend on the species. In canaries (*Serinus canaria*), for example, adult females rather disfavor their foster father's song, which likely helps the individual avoid inbreeding [24]. On the other hand, it can be speculated that learning the father's song as a sexually attractive signal may help individuals of other species avoid outbreeding or hybridization [37–40]. Whether females of a species favor or disfavor incestuous signals might depend on the dispersal patterns and breeding ecology of that species [20, 41, 42]. Based on field observations that white-rumped munias (*Lonchura striata*), the ancestral wild strain of Bengalese finches, sometimes live in a mixed flock with phylogenetically close heterospecifics [43, 44], preference for a song that resembles the father's song might be a good strategy for accepting own-species males while rejecting males of other species.

To examine if female Bengalese finches are sexually imprinted on their father's song, however, more precise control of rearing conditions is necessary. For example, our experimental design could not exclude the possibility that song preference was genetically inherited from parents to daughters, as we used female finches that were cared for by their genetic parents. In addition, because the subjects were housed with parents for a relatively long time (until about 120 dph), it is also possible that such a long-time interaction with the father led to a particularly robust response selectivity to that song. Regarding song learning in males, one study reported that Bengalese finch juveniles can imitate the song of a tutor they interacted with from 35 to 70 dph [45]. When juveniles have normal auditory and social access to a tutor(s), the basic acoustical structure of song elements imitated from the tutor(s) is already visible and stable at 80–90 dph [29, 46]. Therefore, if females are separated from their family earlier in development and are allowed only a limited period or amount of interaction with their father, the strength of the preference for the father's song may become weaker than what we observed in this study. It has also been reported that juvenile male Bengalese finches can memorize and imitate songs from more than one tutor, including genetically unrelated adult males, if they are housed with multiple tutors [47, 48]. Thus, it is reasonable to expect that a female's song preference is similarly shaped through social interaction with multiple adult males, which is actually the case for zebra finches [11, 49]. To better understand the process of preference development, studies focusing on cross-fostering, manipulation of the period of exposure to

genetic or foster father's song, and the number of adult male song tutors will be essential in the future.

Moreover, although we found that songs with higher similarity to the father's song elicited more CSDs, this tendency was not statistically significant. In testing the effect of similarity on female response to unfamiliar songs, the small number of trials available for analysis was a limitation (Fig 3). One possible reason for such low response rate is that the acoustical similarity of unfamiliar songs to the father's song was low overall. Other studies that tested CSDs in estradiol-treated female Bengalese finches similarly reported that the response frequency was low in general when unfamiliar conspecific or heterospecific songs were used as stimuli [3, 26, 29, 50]. Thus, examining the relationship between song features and female CSD responses using unfamiliar stimuli may be difficult. Therefore, manipulation of song similarity based on familiar and preferred songs, rather than a post-hoc analysis as in this study, may enable a more detailed analysis on the effect of particular song features on song preference. Songs of male siblings who shared the song tutor (the father) with the subject females are another candidate for comparison because their songs should resemble that of their father and sound more natural than artificially manipulated songs. However, one study showed that preference for the father's song does not necessarily generalize to the songs of brothers in female zebra finches [51]. Such preference generalization in the case of female Bengalese finches awaits similar experimental investigation.

Regardless of the developmental mechanism of preference for the father's song, our results together with previous reports of song preference in female Bengalese finches measured by other behavioral responses (e.g., operant conditioning, phonotaxis, and vocalizations) [14, 15] imply the importance of considering individual variations in female song preference in this species. In previous studies, researchers have explored acoustical and temporal song characteristics that are preferred across individuals in female Bengalese finches [14, 52–54]. For example, in a call-back assay experiment where song tempo or pitch were manipulated, the authors found that the majority of females preferred faster songs while changes in pitch were not a good predictor of female response [52]. There are other studies that are especially relevant to the characteristics of Bengalese finch song and its evolutionary process. The Bengalese finch is a domesticated strain of its wild ancestor, the white-rumped munia, and the transition probability of Bengalese finch song elements is known to be more complex than that of munias [29, 55]. Thus, it is hypothesized that sexual selection contributed to an increase in sequential complexity across generations. If this is the case, one would predict that that females should possess a preference for such complexity [29]. However, the results of studies have been mixed: although some females did show greater responses to more complex songs (measured by operant conditioning or nest building behavior), there were individual variations in the preference for or sensitivity to the sequential complexity of song [14, 53, 54]. Individual differences in song preference were also reported in another study [26], but it was not known what factor(s) might explain such variation. Our results here suggest that individual differences in developmental song experience in a laboratory breeding environment and/or genetic background may account for this issue. We propose that consideration of the early-life social environment of females is important in re-interpreting previous findings and designing new experiments.

## Supporting information

**S1 File. Effect of E2 implantation on CSD responses to song playbacks.** A brief report of methods and results of a preliminary examination of the hormonal treatment.
(DOCX)

**S2 File. Additional information on song stimuli.** A summary of the number of times each stimulus was presented to each subject.
(DOCX)

**S3 File. Response frequency of each bird.** The total number of CSD occurrences in each test throughout the experiment is shown for each bird.
(DOCX)

## Acknowledgments

We thank Dr. Maki Ikebuchi for assistance in bird breeding, and Dr. Chihiro Mori for instruction of hormone implantation. We are grateful to Dr. Beth A. Vernaleo for her careful proof reading of this paper.

## Author Contributions

**Conceptualization:** Tomoko G. Fujii, Kazuo Okanoya.

**Data curation:** Tomoko G. Fujii.

**Formal analysis:** Tomoko G. Fujii.

**Funding acquisition:** Tomoko G. Fujii, Kazuo Okanoya.

**Methodology:** Tomoko G. Fujii.

**Resources:** Kazuo Okanoya.

**Supervision:** Kazuo Okanoya.

**Visualization:** Tomoko G. Fujii.

**Writing – original draft:** Tomoko G. Fujii.

**Writing – review & editing:** Kazuo Okanoya.

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
