## [Decision Letter · Decision Letter 0]

29 Jul 2021

PONE-D-21-19382

Female Bengalese finches recognize their father’s song as sexually attractive

PLOS ONE

Dear Dr. Fujii,

Thank you for submitting your manuscript to PLOS ONE. After careful consideration, we feel that it has merit but does not fully meet PLOS ONE’s publication criteria as it currently stands. Therefore, we invite you to submit a revised version of the manuscript that addresses the points raised during the review process.

In particular, I agree with the reviewers that it is important to discuss the scope of interpretation based on the limited rearing regimes in the study.  Reviewer 2's comment about the nature of the rearing conditions and external validity are important to discuss, as well as the potential for genetic (experience-independent) contributions to preferences.  I also agree that details about the tutoring regime should be included in the title and abstract and that more of the Introduction and Discussion should be devoted to contextualizing your findings into the broader literature.  

We look forward to receiving your revised manuscript.

Kind regards,

Jon T Sakata, PhD

Academic Editor

PLOS ONE

2. To comply with PLOS ONE submissions requirements, in your Methods section, please provide additional information on the animal research and ensure you have included details on (1) methods of sacrifice, (2) methods of anesthesia and/or analgesia, and (16) efforts to alleviate suffering.

3. In your Methods section, please include a comment about the state of the animals following this research. Were they euthanized or housed for use in further research? If any animals were sacrificed by the authors, please include the method of euthanasia and describe any efforts that were undertaken to reduce animal suffering.

Reviewers' comments:

Reviewer's Responses to Questions

**Comments to the Author**

1. Is the manuscript technically sound, and do the data support the conclusions?

Reviewer #1: Yes

Reviewer #2: Partly

2. Has the statistical analysis been performed appropriately and rigorously? 

Reviewer #1: Yes

Reviewer #2: No

3. Have the authors made all data underlying the findings in their manuscript fully available?

Reviewer #1: Yes

Reviewer #2: No

4. Is the manuscript presented in an intelligible fashion and written in standard English?

Reviewer #1: Yes

Reviewer #2: Yes

5. Review Comments to the Author

Reviewer #1: I enjoyed reading the manuscript “Female Bengalese finches recognize their father’s song as sexually attractive.” In this study, the authors aimed to test whether female Bengalese finches find their father’s song more sexually attractive than unfamiliar songs. In agreement with this hypothesis, playing father’s song elicited a higher number of copulation solicitation responses than playing the songs of unfamiliar adults. The result shows strong evidence of a link between preference for father song, which has been known for decades in finches, and sexual imprinting. The study is well designed and executed, and the manuscript is clearly written. I have only minor comments that I hope will help the authors improve an already good manuscript.

Line 26: As pointed out in the discussion, given that the females were reared by their genetic fathers, this study did not look at the role of early-life experience on the development of preference for father’s song.

Lines 48-49: At least one of the studies cited, Anderson (2009), showed that female swamp sparrows exhibit more CSDs to songs from their local population. Thus, that study also links preference to sexual responses and does not only show “general selectivity to a familiar stimulus”. The main difference I see between the current study and that of Anderson is the identity of the familiar songs to which females show preferential sexual responses (songs from native population vs. songs of fathers).

Lines 48-49: I do not understand what the authors mean by “sexual response in the context of mate choice”. Do they mean sexual responses in general, without considering preference? Or that their study is outside the context of mate choice? Though the birds in their study were not asked to choose a mate, the findings can have implications for mate choice, as pointed out by the authors in the discussion.

Lines 71-74: It seems from reading these lines that at age 120 dph, the females were separated from any males. However, in Figure 1 and elsewhere in the methods it is mentioned that the females were separated from males just a few days before the experiment (about 279 dph). Can the authors please clarify what type of contact (either social or acoustic) did the females have with males between 120 dph and the few days before the experiment? In addition, some other details were not clear to me. Were there at any point, after 120 dph, males in the same room which the females could hear? Were the fathers in the same room?

Line 112: “one subject’s father” – does that mean that the father of every female was used as an unfamiliar bird for other females?

Line 169: I do not see any reason why experience-independent preference for father song would not result in a correlation between preference and similarity to father’s song.

Line 237: Could the lack of statistical significance be due to unfamiliar songs being not that similar to father’s song? (maximum similarity seems just a bit above 60%).

Line 282: The authors may want to compare their results to those of Riebel and Smallegange (2003; ournal of Comparative Psychology, 117(1), 61–66), who found that female zebra finches who have a preference for father’s song do not prefer sibling’s song, despite being similar.

Figure3: The data used to generate these figures have not been disclosed.

Reviewer #2: This manuscript investigates song preference development in female Bengalese f¬inches an important avian model of song and song preference learning and as such of wide interest to all studying plasticity of animal communication signals.

The study investigated adult song preferences of 10 female Bengalese finches from 8 different broods. All females had been raised in small brood cages with their parents and siblings and had been housed under these conditions until they were adult (=120 days posthatching), meaning their father was the only adult song model available.

As young adults, females received estradiol implants to increase behavioural responsiveness - and copulations solicitation displays (CSDs) in particular- to song playback in absence of a live male. Females were then observed during repeated song playback trials and trials were scored to be w/o CSDs. The songs consisted of different songs recorded in the colony and were classified as unfamiliar if the birds had not heard them up until testing and for all females also the father’s song. The authors report that their fathers’ songs – the only song the females heard until day 120 – were soliciting the highest number of trials with copulation solicitation displays.

This is an interesting but also difficult to interpret finding – given that such a strong (almost exclusive) sexual preference for fathers’ song is highly unusual. As discussed below, this finding should perhaps be discussed more in the context of the socially unusual rearing situation. It seems also important that the authors acknowledge and discuss the lack of a control group of females raised in a different social setting to make sure that this is indeed a feature of Bengalese Finch preference development rather than a specific outcome of a specific laboratory rearing condition.

General comments and queries

Could you provide more background information on

1) the availability of song models in normal upbringing – both in the ancestral white-backed munia and in group housed domesticated Bengalese Finches? This would greatly help readers to interpret the findings and form an opinion on the question of whether

- the strong preference for the father’s song is an artefact of the rearing conditions or whether this would this happen with other rearing conditions as well.

- what is known about the sensitive phase for song memorisation in males and females in this species? And when would they normally leave the family group?

- earlier work on this species is poorly cited and discussed – there is substantial work on cross fostering and the development of mating preferences in this species that is not cited here but might shed light on when preferences develop?

2) Should there have been a control group where females are exposed to more social song tutors in the form of a more social housing or the possibility to socialise/disperse at an age where this would normally happen?

3) Looking at the results and in more detail on the data file suggests that overall responsiveness was low overall – but interpreting the data is difficult as the copulation solicitation displays were coded binary, i.e., we do not see the number of CSD given, only whether within a trial a female did or didn’t show any CSDs. Why was this coding chosen and is the father still the most preferred if the absolute number of CSDs for particular songs are analysed instead of the number of trials with CSDs? Do the number of CSDs observed here compare in their magnitude to what other studies in this species reported during song playback tests?

4) A balanced discussion of the advantages of lab tutoring studies (high stimulus control, excellent approach for finding out about learning mechanisms and sensitive phases) versus their disadvantage (less well suited to find out how learning takes place in a socially and spatially more complex environment) should put the results in perspective. Other species might be informative here too: In zebra finches, another well studied estrildid finch, preference development (song learning and visual imprinting) seems to depend highly on social learning -here sufficient lab experiments have been conducted to suggest that there is a sensitive phase for acoustic and visual preference learning/imprinting but that learning can take place with the father as model if there is nobody else but is equally strong if other models are offered during that period.

5) In view of the above I recommend adjusting the current title “Female Bengalese finches recognize their father’s song as sexually attractive” as for one it is semantically ambiguous: ‘they recognise” could mean that “they recognise it as sexually attractive” (leaving it unclear whether this is attractive to them or to other receivers) or that it is sexually attractive to them.

Second, given the restricted social environment with the father as only adult song model up until 120 days the title should contain information indicating the context of a specific experimental rearing situation (at least hinting at the possibility that it is unclear whether the preference for the father’s song observed in this experiment could be an artefact arising from channelled learning mechanisms not receiving the input needed for normal development).

More specific comments by line number:

30 ‘they can perceive..’ perhaps rephrase as it reads like a special talent but females are the intended audience of the species-specific mating signals!?

31 ‘may change’?

110 5 different songs: is this 5 different songs for each of the 10 females (i.e., 50 stimulus songs?) or did some females get the same test songs?

112 ff Was each ‘father’ song used equally often used as unfamiliar control song? Please specify exactly how many test songs there were in total, how many of them were used with how many females and whether all fathers’ song were used as ‘unfamiliar songs’ as well (if this is yielding lengthy text, perhaps a (supplementary) table could summarise this?)

75ff can you give more information on the males – how were they reared? Were any of them siblings/raised by the same tutor?

112 one father or each father?

116 microphone was connected to..?

117 is this the soundcard or the recorder?

119 when were songs recorded? Age? Before (days) or after breeding event?

121 please provide also sample size and range

122 please specify – if you normalise the song by the standard deviation of the waveform (which one? The songs own waveform?) then each song will have another amplitude?

128 please give the settings of the sound level meter

137 does this means that all other females were caught before another was tested? How often were females caught and moved in total during the test period? And with how many days in between catching/moving events?

143/44 please give the number of trials excluded (total, average per female/s.d.) but what was the rationale to not include all tests in the analyses? A female might not show any CSD’s because she was not responsive but it might also be an indication that she found none of the songs attractive enough?

162 Even if there was individual variation: could you give the criteria you used to label females’ behaviour as a CSD, if not replication or comparison with other data becomes difficult.

191 Females were tested 10 times in a row so this variable has a time and order aspect, so should this not be a covariate? If it is analysed as a random variable you lose the information on the order (progress of time of testing) but it is likely that females’ responsiveness increased (if they got better acclimatized or if song further stimulated the reproductive system) or that it decreased if they started to dishabituate

219 was there an interaction between song type and presentation order?

Table 1: presentation order is here 1-5, but there were 10 test days, could you clarify this?

And there is also a random variable test number with 9 levels. Have you now not coded the same variable in two ways? I think the model should only contain the test day or test order – with so few birds and so many factors this model seems potentially overfitted?

259/60 in view of the rearing conditions (father as only model until day 120) being so different from a normal upbringing, I suggest to phrase this conclusion more carefully

267 how would selection take place on the population level – the fitness advantage of individual mating preferences is on the individual level?

287 for the benefit of the reader: please specify what the previous findings in ref 14 & 15 were and how they support your argument.

6. PLOS authors have the option to publish the peer review history of their article (what does this mean?). If published, this will include your full peer review and any attached files.

Reviewer #1: No

Reviewer #2: No

---

## [Author Response · Author response to Decision Letter 0]

8 Sep 2021

General comments:

We appreciate the reviewers’ and editor’s thoughtful and helpful comments on our manuscript. We acknowledge that our subject females may have developed under unusual social conditions due to a unique laboratory setting. As pointed out by the reviewers and the editor, the rearing regime should be clearly specified in the text, and the results must be interpreted more carefully. Thus, we revised the introduction, discussion, and related sentences in other sections to clarify the limitations of the current study as well as its contributions to the current body of research. Specific responses to each comment and the modifications made are described below. Original comments and questions from the reviewers/editor are italicized. All page and line numbers which specify the location of changes refer to the marked-up version of our revised manuscript unless otherwise mentioned.

Academic editor:

> In particular, I agree with the reviewers that it is important to discuss the scope of interpretation based on the limited rearing regimes in the study. Reviewer 2's comment about the nature of the rearing conditions and external validity are important to discuss, as well as the potential for genetic (experience-independent) contributions to preferences. I also agree that details about the tutoring regime should be included in the title and abstract and that more of the Introduction and Discussion should be devoted to contextualizing your findings into the broader literature. 

 Thank you for the important suggestions. According to the comment, we additionally discussed the limitations that may come from the unusual tutoring conditions (p. 19, lines 312-313; pp. 20-21, lines 334-347) and included rearing details in the title and abstract. We also revised the introduction and discussion sections to explain the purpose (p. 5, lines 62-67) and results of our study in relation to the previous findings reported in the broader literature (pp. 19-20, lines 324-328). We hope that our description now provides a sufficient amount of detail. Please also refer to our responses to similar comments raised by reviewer 2 (pp. 4-5 of this letter).

> 1. Please ensure that your manuscript meets PLOS ONE's style requirements, including those for file naming.

 We ensured that the format and style meet PLOS ONE’s journal requirement.

> 2. To comply with PLOS ONE submissions requirements, in your Methods section, please provide additional information on the animal research and ensure you have included details on (1) methods of sacrifice, (2) methods of anesthesia and/or analgesia, and (16) efforts to alleviate suffering.

(1) No animals were sacrificed by the authors during the course of this research. (2) We used local anesthesia for E2 tube implantation. The type of anesthesia and surgical procedures are described in (p. 8, line 124). (16) While revising the methods section in response to reviewer 2’s comment, we added a description of how birds were handled during the experiment (p. 10, lines 165-169).

> 3. In your Methods section, please include a comment about the state of the animals following this research. Were they euthanized or housed for use in further research? If any animals were sacrificed by the authors, please include the method of euthanasia and describe any efforts that were undertaken to reduce animal suffering.

 After this study, all birds continued to be housed in our laboratory colony for additional research (p. 6, lines 100-102), and no animals were sacrificed.

> 4. Please include captions for your Supporting Information files at the end of your manuscript, and update any in-text citations to match accordingly.

 We added the list and captions for the Supporting information files at the end of our manuscript and checked the in-text citations.

Reviewer 1:

> Line 26: As pointed out in the discussion, given that the females were reared by their genetic fathers, this study did not look at the role of early-life experience on the development of preference for father’s song.

 As you pointed out, the role of early experience remains unclear until preference is tested in cross-fostering experiments. We modified the sentence so that it focuses on future directions rather than suggesting an experience-dependent mechanism for the current results (p. 2, lines 28-31).

> Lines 48-49: At least one of the studies cited, Anderson (2009), showed that female swamp sparrows exhibit more CSDs to songs from their local population. Thus, that study also links preference to sexual responses and does not only show “general selectivity to a familiar stimulus”. The main difference I see between the current study and that of Anderson is the identity of the familiar songs to which females show preferential sexual responses (songs from native population vs. songs of fathers).

 Thank you. It might be unfair that we did not refer to Anderson (2009) here, as it already demonstrated the correlation between operant responses and CSDs at least when female swamp sparrows were tested with local/foreign song dialects. We added this reference and another related study (Holveck & Riebel, 2007) and changed the description accordingly (p. 4, lines 50-56).

> Lines 48-49: I do not understand what the authors mean by “sexual response in the context of mate choice”. Do they mean sexual responses in general, without considering preference? Or that their study is outside the context of mate choice? Though the birds in their study were not asked to choose a mate, the findings can have implications for mate choice, as pointed out by the authors in the discussion.

 We intended to convey that, strictly speaking, song preference can only be interpreted as mate choice if the ecological validity of the response is justified by other behavioral indices (such as CSDs). We agree that the previous description was ambiguous and modified the sentence for clarity (p. 4, lines 52-53).

> Lines 71-74: It seems from reading these lines that at age 120 dph, the females were separated from any males. However, in Figure 1 and elsewhere in the methods it is mentioned that the females were separated from males just a few days before the experiment (about 279 dph). Can the authors please clarify what type of contact (either social or acoustic) did the females have with males between 120 dph and the few days before the experiment? In addition, some other details were not clear to me. Were there at any point, after 120 dph, males in the same room which the females could hear? Were the fathers in the same room?

 We agree that the social/acoustic environment of the females after separation from their family was not clear enough in the original manuscript. From 120 dph to the beginning of the experiment, females were kept in a single-sex cage, but conspecific males were also kept in other cages placed in the same room. Thus, during this period, the females could hear and see males although they could not physically interact with them. However, we made sure that their fathers were moved to another room so that the subjects had no acoustical and social access to their fathers between120 dph and the start of the experiment. We added this description in the methods section (p. 6, lines 86-90).

> Line 112: “one subject’s father” – does that mean that the father of every female was used as an unfamiliar bird for other females?

 We meant that each subject’s father’s song was presented to other subjects as an unfamiliar song. We changed the description to provide clarity (p. 8, line 134). Please also see our comment to a similar question from reviewer 2 (pp. 8-9 in this letter).

> Line 169: I do not see any reason why experience-independent preference for father song would not result in a correlation between preference and similarity to father’s song.

 As you pointed out, the description was not appropriate because the same prediction can be made regardless of the mechanisms underlying preference for the father’s song in females (either genetic or experience-dependent). We changed the sentence to remove the mention of experience-dependent developmental mechanisms here (p. 12, lines 208-209). We also changed the term in another sentence where we irrelevantly used ‘experience’ (p. 8, line 136).

> Line 237: Could the lack of statistical significance be due to unfamiliar songs being not that similar to father’s song? (maximum similarity seems just a bit above 60%).

 Yes, that possibility cannot be excluded. We added this point in the discussion section (p. 21, lines 351-352). Thank you for your suggestion.

> Line 282: The authors may want to compare their results to those of Riebel and Smallegange (2003; Journal of Comparative Psychology, 117(1), 61–66), who found that female zebra finches who have a preference for father’s song do not prefer sibling’s song, despite being similar.

 While answering your question above (to line 237 in the original manuscript), we also cited Riebel & Smallegange (2003) to indicate the possibility of using the brother’s song in future studies (p. 21, lines 357-362).

> Figure3: The data used to generate these figures have not been disclosed.

 We uploaded all the data used to generate figures to the figshare depository. For Fig. 3, we used the dataset named ‘Fujii&Okanoya_PONE_Dataset2 (song similarity).xlsx’. It is available from the URL below. https://figshare.com/s/dee039e4ac403fbd2a7e (doi: 10.6084/m9.figshare.14677572)

Reviewer 2:

> This is an interesting but also difficult to interpret finding – given that such a strong (almost exclusive) sexual preference for fathers’ song is highly unusual. As discussed below, this finding should perhaps be discussed more in the context of the socially unusual rearing situation. It seems also important that the authors acknowledge and discuss the lack of a control group of females raised in a different social setting to make sure that this is indeed a feature of Bengalese Finch preference development rather than a specific outcome of a specific laboratory rearing condition.

 Thank you very much for the important suggestions. We acknowledge the limitation of our study that the preference for father’s song found here cannot be generalized without testing birds reared in more natural settings with multiple social tutors and/or shorter periods of interaction with the tutor(s). Thus, we revised the discussion section (mainly in pp. 20-21, lines 334-347) and the related sentences in other sections so that the results can be interpreted more carefully.

Please let us clarify our standpoint to ensure correct communication hereafter: Our primary purpose was to test whether the preference to father’s song, if any, is sexually motivated. We do not intend to show that such a strong preference for father’s song is a general phenomenon that is expected to occur in this species under any rearing conditions. It has already been shown with other behavioral indices that female adult Bengalese finches (and zebra finches) respond more to their (foster) father’s song compared to unfamiliar conspecific songs (Kato et al., 2010; Fujii et al., 2021). As it was previously unclear whether this preference can be interpreted as a sexual preference for a male who sings the song, we designed the current study. In the previous manuscript, we stated that ‘Bengalese finch females prefer their father’s song’ because we thought we could not disentangle the genetic and experience-dependent effect on the development of that preference. We rather believe that the early-life social/auditory experience with adult male(s) should be critical to the development of song preference, and thus mating preference would be affected by multiple songs that females are exposed to early in life, if multiple male tutors are available. Through revising the manuscript, including the title and abstract, we attempted to disambiguate these points.

> Could you provide more background information on:

1) the availability of song models in normal upbringing – both in the ancestral white-backed munia and in group housed domesticated Bengalese Finches? This would greatly help readers to interpret the findings and form an opinion on the question of whether

- the strong preference for the father’s song is an artefact of the rearing conditions or whether this would this happen with other rearing conditions as well.

- what is known about the sensitive phase for song memorisation in males and females in this species? And when would they normally leave the family group?

- earlier work on this species is poorly cited and discussed – there is substantial work on cross fostering and the development of mating preferences in this species that is not cited here but might shed light on when preferences develop?

We appreciate the reviewer’s careful reading and helpful suggestions. In the previous manuscript, we did not provide enough information on the ecology of the Bengalese finch and white-rumped munias, including song learning processes. As you suggested, we cited additional references about the song learning process in male juvenile Bengalese finches (Clayton, 1987; Soma et al., 2009; Takahasi et al., 2010; Tachibana et al., 2017) and preference development in female zebra finches (Clayton, 1988; Holveck & Riebel, 2014) in the discussion section (pp. 20-21, lines 334-347). To our knowledge, however, there are very few studies that investigated the development of mating preference in female Bengalese finches. Although researchers have conducted many cross-fostering experiments in songbirds since the late 20th century, Bengalese finches were usually used as foster parents of other species such as zebra finches. Also, it is difficult to specify the exact age of independence and availability of song tutors in wild white-rumped munias. Although knowledge of their ecology is limited, we provided a little more information about the social environment of this wild species (pp. 19-20, lines 324-328). We hope that this elaboration with additional references on laboratory studies in male Bengalese finches and other species of females aids in interpreting the findings of the current study.

> 2) Should there have been a control group where females are exposed to more social song tutors in the form of a more social housing or the possibility to socialise/disperse at an age where this would normally happen?

 We do agree that rearing female birds in a more natural setting is an essential step in order to elucidate how song preference develops through experience. However, we believe this is outside the scope of our experiment, as the current study only examined whether the preference for father’s song can be interpreted as mating preference. Thus, we revised the manuscript so that the discussion section focuses more on the study limitations and the importance of future studies to approach these remaining issues (pp. 20-21, lines 334-347). In addition, the housing conditions in the current study (keeping family members together until the juveniles reach 120 dph) were set to match the conditions used in the previous studies which reported a preference for the father’s song in female Bengalese finches using operant conditioning or phonotaxis (Kato et al., 2010; Fujii et al., 2021). We also explained this in the introduction and method sections (p. 4, lines 66-67 & p. 5, 81-82).

3) Looking at the results and in more detail on the data file suggests that overall responsiveness was low overall – but interpreting the data is difficult as the copulation solicitation displays were coded binary, i.e., we do not see the number of CSD given, only whether within a trial a female did or didn’t show any CSDs. Why was this coding chosen and is the father still the most preferred if the absolute number of CSDs for particular songs are analysed instead of the number of trials with CSDs? Do the number of CSDs observed here compare in their magnitude to what other studies in this species reported during song playback tests?

 Thank you for your question and comment. The reason we originally only used binary coding was that it was difficult to compare the number of CSDs due to individual variations in the duration of each CSD bout (please refer to the movies uploaded to the figshare repository for examples). Thus, in addition to the response frequency, we decided to analyze the total duration of CSDs during song presentation and compared the duration between the father’s song and unfamiliar songs for 7 out of 10 birds that responded to at least one unfamiliar song (method: p. 12, lines 194-196 & 201-206, result: p. 15, lines 260-265 and Fig. 2 (b)). Most birds (6 out of 7) showed longer displays to the father’s song than to unfamiliar songs although this tendency was not statistically significant. We believe this additional analysis along with response frequency data provides greater insight into song preference than frequency data could alone.

Please also note that although the overall response frequency was low, the response rate to the father’s song was relatively high: the response rate to the father’s song was the highest in 10 birds, and 4 out of them exhibited CSD(s) every time they were exposed to the father’s song. This suggests a high selectivity of response even when analyzed with frequency alone. As we explain in p. 10 in this letter, no trials were excluded from any statistical analysis or figure illustrations. The response rate reported in the text and illustrated in Fig. 2 (a) was calculated from all trials we conducted.

 For the last part of your question, we think that the response frequency or magnitude of CSDs in our subjects was comparable to the results of similar studies. For example, Clayton & Pröve (1989) recorded CSDs during 12 minutes of song presentation in E2-implanted female Bengalese finches. They reported that the population mean of the total CSD frequency was about 20 times for conspecific song presentation, while the mean frequency was much less (~ 5 times) for heterospecific songs. In another study (Dunning et al., 2014), the authors tested E2-implanted females (12 birds) with 8 different unfamiliar conspecific songs for each individual. Although the exact number of songs was not described, they reported that all females responded to at least one song, but many songs evoked no CSDs at all. Although a direct comparison must be done with caution due to differences in conditions and experimental environment, it seems that the frequency of CSDs substantially depends on the type of song stimulus and can be very low particularly when presented with unfamiliar songs. We briefly referred to these previous studies in the discussion to help provide context for our results (p. 21, lines 352-354).

4) A balanced discussion of the advantages of lab tutoring studies (high stimulus control, excellent approach for finding out about learning mechanisms and sensitive phases) versus their disadvantage (less well suited to find out how learning takes place in a socially and spatially more complex environment) should put the results in perspective. Other species might be informative here too: In zebra finches, another well studied estrildid finch, preference development (song learning and visual imprinting) seems to depend highly on social learning -here sufficient lab experiments have been conducted to suggest that there is a sensitive phase for acoustic and visual preference learning/imprinting but that learning can take place with the father as model if there is nobody else but is equally strong if other models are offered during that period.

 We acknowledge that discussing the advantages and disadvantages of laboratory experiments vs. studies in a more natural and complex social environment is very important, and was lacking in our original manuscript. We cited other references, including studies in zebra finches as you suggested, and other studies that investigated Bengalese finch song learning with multiple male tutors (pp. 20-21, lines 334-347).

5) In view of the above I recommend adjusting the current title “Female Bengalese finches recognize their father’s song as sexually attractive” as for one it is semantically ambiguous: ‘they recognise” could mean that “they recognise it as sexually attractive” (leaving it unclear whether this is attractive to them or to other receivers) or that it is sexually attractive to them. Second, given the restricted social environment with the father as only adult song model up until 120 days the title should contain information indicating the context of a specific experimental rearing situation (at least hinting at the possibility that it is unclear whether the preference for the father’s song observed in this experiment could be an artefact arising from channelled learning mechanisms not receiving the input needed for normal development).

 We changed our title to avoid any ambiguity and to include more information on the rearing conditions of our subjects.

More specific comments by line number:

> 30 ‘they can perceive..’ perhaps rephrase as it reads like a special talent but females are the intended audience of the species-specific mating signals!?

> 31 ‘may change’?

 It might be incorrect or misleading to say that female birds ‘perceive’ song variations. We revised this sentence to ‘they are sensitive to inter- and intra-species variations and may change their behavior…’ so that the message is more correctly conveyed to the readers (p. 3, lines 34-35).

> 110 5 different songs: is this 5 different songs for each of the 10 females (i.e., 50 stimulus songs?) or did some females get the same test songs?

 All subjects were tested with a unique stimulus set, but there was some overlap between the songs used for each female. (Please see our response to your comment to line 112 for details.)

> 112 ff Was each ‘father’ song used equally often used as unfamiliar control song? Please specify exactly how many test songs there were in total, how many of them were used with how many females and whether all fathers’ song were used as ‘unfamiliar songs’ as well (if this is yielding lengthy text, perhaps a (supplementary) table could summarise this?

 We acknowledge that more detailed information about these song stimuli is necessary and helpful for readers. As you suggested, we listed all the song stimuli used for each female in a table as a part of supporting information S2 File and updated the text (p. 8, lines 134-137). The total number of songs used as test stimuli is identical to the number of males used to record the song stimuli (19 birds), which is also shown in the Methods subsection ‘Animals’ (p. 6, lines 91-92).

> 75ff can you give more information on the males – how were they reared? Were any of them siblings/raised by the same tutor?

 We added the rearing and family conditions of these males (as far as we know, since some birds were purchased from breeders). None of the birds were siblings, or tutored by the same male, which means all males used in these experiments had distinct songs and did not share song features (p. 6, lines 92-97).

> 112 one father or each father?

We meant ‘each father.’ We clarified it in the sentence (p. 8, line 134). Please also see the Supporting Information S2 File and our comments to your questions to lines 110 and 112 above.

> 116 microphone was connected to..?

> 117 is this the soundcard or the recorder?

 The signals recorded from the microphone were processed through an amplifier and soundcard, then stored onto a PC. We changed the description to make it clear (p. 9, lines 141-143).

> 119 when were songs recorded? Age? Before (days) or after breeding event?

 We added a bit more information about the timing of song recording in the Methods section (pp. 8-9, lines 138-141). At the time we performed song recordings, all the males (singers) were adult birds (> 180 dph). Because we could not tell the exact age of the males that were purchased from a breeder, we can only provide this lower limit. Also, songs of the subjects’ fathers were recorded before they were used for breeding. It is well known that the song features of Bengalese finches do not change depending on season, age or breeding experience, once the song is crystallized at about 120 dph (Okanoya & Yamaguchi, J Neurobiol, 1997; Tachibana, Koumura & Okanoya, J Comp Physiol A, 2015).

> 121 please provide also sample size and range

 We added the sample size (as we used 6 renditions for each song, the total number of renditions were 6 * 19 = 114) and the rage of the stimulus duration (p. 9, lines 146-148). We also included the same information on the total duration of the 6 renditions for each singer.

> 122 please specify – if you normalise the song by the standard deviation of the waveform (which one? The songs own waveform?) then each song will have another amplitude?

 The procedure was used to normalize the amplitude within one song (among the 6 renditions) and also between different songs. We added this description (p. 9, lines 150-151).

> 128 please give the settings of the sound level meter

 We added the time weighting settings (‘fast’) at the end of the sentence (p. 9, line 156-157). The frequency weighting was set to type A, which is also specified in this sentence (p. 9, line 154).

> 137 does this means that all other females were caught before another was tested? How often were females caught and moved in total during the test period? And with how many days in between catching/moving events?

 Yes, we handled the other females when moving them to the other soundproof chamber before testing a subject female. When moving them back to the cage after testing, we allowed the birds to fly back into the test cage without handling them. As we tested 1 bird or 2 birds in the morning of a day and each bird was tested once in 2 days, each bird was handled once or twice per day during the test period (ranged from 10 to 18 days). We added this information in the manuscript (p. 10, lines 165-169).

> 143/44 please give the number of trials excluded (total, average per female/s.d.) but what was the rationale to not include all tests in the analyses? A female might not show any CSD’s because she was not responsive but it might also be an indication that she found none of the songs attractive enough?

 We did not exclude any trials from the analysis or the figure illustration (as written in the original manuscript p. 10, lines 163-164). We realized that it might appear as if we excluded the ‘invalid’ trials if we use the terms ‘valid/invalid.’ To avoid confusion, we changed the sentences that included these terms (pp. 10-11, lines 175-178).

> 162 Even if there was individual variation: could you give the criteria you used to label females’ behaviour as a CSD, if not replication or comparison with other data becomes difficult.

 We identified the behavior as a CSD if a female showed all the following movement features: tilted head and chest down, tail raised and quivering. We revised this part of the manuscript to so that the criteria are clearer and easier to understand (p. 11, lines 191-193).

> 191 Females were tested 10 times in a row so this variable has a time and order aspect, so should this not be a covariate? If it is analysed as a random variable you lose the information on the order (progress of time of testing) but it is likely that females’ responsiveness increased (if they got better acclimatized or if song further stimulated the reproductive system) or that it decreased if they started to dishabituate

 As you pointed out, the progress of time during the experiment might increase the response due to acclimation to the environment or decrease the response due to habituation to the stimuli. However, it is also possible that these effects interact or that there are individual differences in how these effects work. The data actually show that the number of tests required to reach the criterion of 5 responses varied between individuals. Thus, we found it difficult to assume a simply positive or negative effect of time on the response frequency. For those reasons, we decided to include this variable as a random effect rather than a fixed effect.

> 219 was there an interaction between song type and presentation order?

 Although we did not include the interaction between song type (unfam/father = 0/1) and presentation order (1~5) in the original GLMM analysis, it may be reasonable to assume such an interaction. Here, we plotted the response frequency against either the song type or presentation order with logistic regression curves. Two figures show exactly the same data in 2 different ways. As you can see in the figures, it can be speculated that the effect of song type is more significant in later trials than in earlier trials, or in other words, the effect of presentation order is more significant in trials of unfamiliar songs than the father’s song. 

Thus, we ran another GLMM analysis, in which all parameters were the same as those written in the original manuscript except that the interaction term (song type x presentation order) was additionally introduced. The estimated parameters are shown in the table below. However, the model estimation resulted in a singular fit, probably due to our limited samples. Because the primary purpose of this model analysis is to statistically describe the effect of song type, and the effect related to presentation order is an inevitable artifact due to experimental design, we think that a simpler, non-overfitted model (the original one) better suits this purpose rather than a more complex model that may lack generality.

Independent variables Estimates 　 z-value p-value

Fixed effects coefficient s.e. 　 　

 intercept -0.991 0.682 　 　

 song type (unfam/father = 0/1) 2.287 0.928 2.464 0.014 

 presentation order (1-5) -0.770 0.245 -3.140 0.002 

 interaction (song type × order) 0.857 0.331 2.585 0.010 

Random effects variance s.d. 　 　

 subject ID (10 levels) 1.064 1.031 　 　

 stimulus ID (19 levels) 0.000 0.000 

 test number (9 levels) 0.208 0.456 　 　

> Table 1: presentation order is here 1-5, but there were 10 test days, could you clarify this?

And there is also a random variable test number with 9 levels. Have you now not coded the same variable in two ways? I think the model should only contain the test day or test order – with so few birds and so many factors this model seems potentially overfitted?

 The presentation order here means the order of 5 stimuli presented within a single test, not the order of the test date. As described in the Methods section, one test was composed of 5 presentations of different song stimuli. Thus, we are sure that the model does not contain the same variable coded in two ways. In constructing the statistical model, we plotted the birds’ response against the variables that possibly affect the response frequency and found that a song stimulus tended to elicit more responses when presented earlier within a test. Thus, we thought that this parameter is not negligible and should be included in the model not to overestimate the effect of stimulus similarity on CSD responses.

> 259/60 in view of the rearing conditions (father as only model until day 120) being so different from a normal upbringing, I suggest to phrase this conclusion more carefully

 We acknowledge that this is an important point. We stated here that the results need careful interpretation and discussed the limitations of the current study in later paragraphs (p. 19, lines 312-313).

> 267 how would selection take place on the population level – the fitness advantage of individual mating preferences is on the individual level?

 We changed the term ‘species’ to ‘individual’ as it was confusing and inappropriate (p. 19, lines 320-322). As you point out, we did not mean that the selection works at the species level but wanted to convey that an individual who disfavors her father’s song may be able to avoid inbreeding.

> 287 for the benefit of the reader: please specify what the previous findings in ref 14 & 15 were and how they support your argument.

 We inserted a brief description about these references as we agree that the previous manuscript was not informative enough for the readers (p. 22, lines 364-365).

---

## [Decision Letter · Decision Letter 1]

25 Oct 2021

PONE-D-21-19382R1Preference for father’s song in female Bengalese finches measured by sexual displays in a laboratory environmentPLOS ONE

Dear Dr. Fujii,

Thank you for submitting your manuscript to PLOS ONE. After careful consideration, we feel that the manuscript continues to have merit but does not fully meet PLOS ONE’s publication criteria as it currently stands. One reviewer only has a single minor request, whereas another reviewer continues to have substantive comments that need to be addressed. Therefore, we invite you to submit a revised version of the manuscript that addresses the points raised during the review process.

We look forward to receiving your revised manuscript.

Kind regards,

Jon T Sakata, PhD

Academic Editor

PLOS ONE

Journal Requirements:

Reviewers' comments:

Reviewer's Responses to Questions

**Comments to the Author**

1. If the authors have adequately addressed your comments raised in a previous round of review and you feel that this manuscript is now acceptable for publication, you may indicate that here to bypass the “Comments to the Author” section, enter your conflict of interest statement in the “Confidential to Editor” section, and submit your "Accept" recommendation.

Reviewer #1: All comments have been addressed

Reviewer #2: (No Response)

2. Is the manuscript technically sound, and do the data support the conclusions?

Reviewer #1: Yes

Reviewer #2: Partly

3. Has the statistical analysis been performed appropriately and rigorously? 

Reviewer #1: Yes

Reviewer #2: No

4. Have the authors made all data underlying the findings in their manuscript fully available?

Reviewer #1: Yes

Reviewer #2: Yes

5. Is the manuscript presented in an intelligible fashion and written in standard English?

Reviewer #1: Yes

Reviewer #2: Yes

6. Review Comments to the Author

Reviewer #1: The authors have addressed my comments satisfactorily. The only remaining suggestion that I have is that they should summarize their findings about the correlation between similarity and preference in the Abstract. Other than that, I wish good luck to the authors.

Reviewer #2: The authors have written a detailed reply taking care to respond to all issues. For many this is satisfactory but a few major and minor points remain to clarify.

1) As the study misses a control group with different song experiences/crossfostering the abstract and title should perhaps better be adjusted to be unambiguous regarding the specific experimental (narrow) context? A (hypothetical, not ideally phrased yet) title like “Female BF show sexual displays towards father song if this was only song model they experienced when juvenile” would be clear in this respect (not these exact wordings but as an indication how the title could indeed summarise the study).

Such an adjusted title would alert readers from the beginning that this experiment might not trace normal development - this is not per se a critique of highly controlled experimental setups, at the contrary they can be very informative regarding learning mechanisms etc, and thus provide insights in the mechanisms of social transmission/learning. However, they provide only limited insights regarding transmission direction/ model choice if the experiment offers no choice with respect to song models. As is, the title suggests insights into a general feature of BF song preference development but with the lack of alternative song models during development you have tested a highly specific case only – readers should be informed about this in the title and abstract.

2) Binary coding of CSDs (reply point 3):

The justification given to code CSDs as a binary variable is that its duration varied and separate vs. long CSD were not always easy recognisable – this suggest that total duration spend in CSD would be a better measure but then you report that duration doesn’t show a difference across song categories. Does that mean, that if there were displays to other songs they must have longer? (as the yes/no occurrence was higher during father playbacks)? Should this noticable difference among unfamiliar/fathers’ songs not be discussed in more detail? Related to this point: the additional references you list here to compare duration/frequency of CSD with other studies should be discussed in more detail (now disc. only mentions these studies) and help readers to understand which measure is chosen most often in the Bengalese finch and why.

3) Glmm analyses: song presentation order within a test is now a fixed factor and the repeated testing days/events (test number) a random factor. This means that you are asking the model i) to check whether each position within the test order is different from any of the other (ignoring that there is an order/time effect from the 1st to 5th position) and ii) that your analyses do not address the question of whether order and stimulus type might show an interaction. Likewise, ‘test’ is now a random factor rather than a covariate meaning that also for this variable you again omit testing of whether order or and temporal effects occur within a test series. In short, neither within a test nor in the test series, is order or time controlled statistically. The current analyses - by treating test as a random variable - will not detect an order or time effect. The reason you give in the reply for not checking for an order/time effect is saying you could not predict the direction of the effect but this argument is irrelevant- you want to test whether repeated testing leads to an effect over time (should this be the case; data inspection can show you whether there was an in- or decrease).

Additional specific comments by line number

20-21 add ‘if the father’s song is the only song they heard’ or mention that you want to know whether it is possible at all? As is you ask for a function of something that might be an artefact of the procedure, the question of function/consequence should not be about this specific song, but whether any song (incl father) heard during this time leads to a preference? (on first principles: if there is preference learning and females learn during a sensitive phase, then if there is only one thing to learn they are likely to learn (cf. ducklings imprinting on footballs) and if these learned preferences normally affect mating preferences this is likely also the case to involve the father’s song?) Here you ask if such preference lead to matings but this has already been shown in Bengalese finches – so you should be specific here saying it is to test whether this would also happen with the father’s song?

27 this conclusion is too strong. Again, be clear about the reductionist environment: to be sure that this statement is correct you first would have to test if this effect also occurs if the father isn’t the only tutor and if females are raised in a more natural/more socially varied environment (as you suggest in the sentences after). Adjust the statement here so it is not general but that this preference is seen in the current circumstances and make clear that to test if a learned preference for the father’s song is relevant for mate choice it should also been seen if other songs were also heard, or should likewise be seen if the females had been exposed to another song (e.g. a foster father, cage mates during the sensitive phase etc..).

37 odd sentence, perhaps rephrase? (Species recognition is not something uniquely special to female songbirds but occurs across species – it is the very essence of a mating signal?)

38 independently

39 These references do not support the statement. The sentence makes a statement about song birds in general and that some species develop experience independent preferences but the quotes are two experimental studies in the zebra finch (the two references thus refer to only one species and this is a species where preference is NOT experience-independent, but learned to a large degree).

47 or another male’s/males’ song? (can be new/different adult male or from peer group)

48ff please check the literature again: there is plenty of evidence for several species (and in particular the zf which you discuss here) that these learned preferences affect mate choice/differential allocation etc.. to say here it has not been tested ignores a lot of previous work (zebra finch, cowbirds, white crowned sparrow, song sparrow, etc.. etc..)

50 again you are only citing two zebra finch studies while making general statements about song birds – what about e.g., Darwin finches as example for pref. for father’s song?

56-58 you can’t address this here – drop?

302 this is misleading – it should be preference for the song(s) they heard early in life – we do not know whether outside the limited exposure in the laboratory (only the father is available as tutor) which model (or several models) are influencing preferences

303 sweeping statement (and not correct) others have tested in both species whether song preferences translate to mate preferences – so please be specific! Perhaps what you want to say is that in BF that if such a preference for the father’s song exist that it will also lead to more CSD? It has definitely been shown for both species that song preferences (measured with a variety of methods) predict live male preferences/mating/pair formation.

316 Nicky Clayton’s work on cross fostering T.g.g and T.g.c. has shown that the females imprint on the father’s subspecies song and choose mates accordingly (the series of experiments is reviewed in (Clayton 1990).

Clayton NS 1990: Assortative mating in zebra finch subspecies, Taeniopygia guttata guttata and T. g. castanotis. Philosophical Transactions of the Royal Society of London Series B-Biological Sciences 330: 351-370. 10.1098/rstb.1990.0205

7. PLOS authors have the option to publish the peer review history of their article (what does this mean?). If published, this will include your full peer review and any attached files.

Reviewer #1: No

Reviewer #2: No

---

## [Author Response · Author response to Decision Letter 1]

27 Dec 2021

General comments:

 Thank you very much again for your careful reading and suggestions. We now revised the manuscript to respond to the major and minor points raised by the reviewers. Because we needed to adjust the number of words in the abstract, we also modified some sentences that were not directly related to the reviewers’ comments, but these changes are also highlighted in the marked-up version. Specific responses and details about our modifications are described below. Line numbers refer to the marked-up version of our revised manuscript, unless otherwise noted.

> We added a URL for one reference [31], which was lacking in the previous version. As we cited one additional reference [34] and changed the order of citations, the reference numbers from 16 to 53 differ from the previous manuscript. Other than that, the references remain the same.

Reviewer #1:

The authors have addressed my comments satisfactorily. The only remaining suggestion that I have is that they should summarize their findings about the correlation between similarity and preference in the Abstract. Other than that, I wish good luck to the authors.

> We appreciate your comments in the previous and current round of the review. In the latest version, we revised the abstract so that it also refers to the results regarding the correlation between stimulus similarity and preference (p. 2, lines 29-30).

Reviewer #2:

The authors have written a detailed reply taking care to respond to all issues. For many this is satisfactory but a few major and minor points remain to clarify.

> Thank you again for your very helpful comments and suggestions. We acknowledge the importance of precise communication, especially regarding the research purpose and the conclusions inferred by the results. We also understand the need to carefully explain the specific rearing conditions in the current study. Please see the following responses to each comment.

1) As the study misses a control group with different song experiences/crossfostering the abstract and title should perhaps better be adjusted to be unambiguous regarding the specific experimental (narrow) context? A (hypothetical, not ideally phrased yet) title like “Female BF show sexual displays towards father song if this was only song model they experienced when juvenile” would be clear in this respect (not these exact wordings but as an indication how the title could indeed summarise the study). Such an adjusted title would alert readers from the beginning that this experiment might not trace normal development - this is not per se a critique of highly controlled experimental setups, at the contrary they can be very informative regarding learning mechanisms etc, and thus provide insights in the mechanisms of social transmission/learning. However, they provide only limited insights regarding transmission direction/ model choice if the experiment offers no choice with respect to song models. As is, the title suggests insights into a general feature of BF song preference development but with the lack of alternative song models during development you have tested a highly specific case only – readers should be informed about this in the title and abstract.

> We appreciate your suggestion for the title. While revising the manuscript, however, we felt that your suggested title might unnecessarily direct the readers towards a very limited interpretation of the manuscript. As we clearly described the research purpose and the limitations of the study in the revision, we felt a slightly broader title was warranted. Please let us clarify our points here again: It has already been shown in previous studies that female Bengalese finches express selective phonotaxis, vocalizations, or operant behaviors to their father’s song, when they were reared in a single family and exposed to the father’s song almost exclusively. Though this differs from a wild environment, it is a standard social condition for Bengalese finches, as they are a domesticated species. However, song preference measured by such general behavioral indices (such as phonotaxis mentioned above) does not necessarily mean that females are sexually attracted to the father’s song. As mentioned in the introduction, there are indeed some studies that showed the correlation between CSDs and other responses to male songs, but it is unclear if this is also the case for the father’s song specifically. We believe that our study provides important insights for interpreting the previous results and designing future research, as laboratory studies of song preference usually use subjects reared under similar conditions to our study. Thus, the preferences of such birds can be substantially influenced by developmental song exposure. Of course, it is unlikely that a strong preference for the father’s song would generally occur in birds reared in a more complex social environment, though such a situation is rather unusual for a domesticated species such as the Bengalese finch. We acknowledge that this is out of the current study’s scope and thus discussed the limitations in the abstract and the discussion section. In summary, in this paper, we do not argue that female BFs generally develop a strong preference for the father’s song, but would like test if such a preference can be interpreted as a sexual preference. Therefore, we would like to use the title (full & short) as follows:

Full: Female Bengalese finches show selective sexual displays to their father’s song

Short: Selective sexual displays to the father’s song in female Bengalese finches

2) The justification given to code CSDs as a binary variable is that its duration varied and separate vs. long CSD were not always easy recognisable – this suggest that total duration spend in CSD would be a better measure but then you report that duration doesn’t show a difference across song categories. Does that mean, that if there were displays to other songs they must have longer? (as the yes/no occurrence was higher during father playbacks)? Should this noticable difference among unfamiliar/fathers’ songs not be discussed in more detail? Related to this point: the additional references you list here to compare duration/frequency of CSD with other studies should be discussed in more detail (now disc. only mentions these studies) and help readers to understand which measure is chosen most often in the Bengalese finch and why.

> There have been only a few studies in Bengalese finches that measured CSDs (Clayton & Pröve, 1989; Okanoya, 2004; Dunning et al., 2014). Clayton & Pröve and Dunning et al. analyzed the frequency of CSDs, defined as the number of display bouts, while Okanoya reported whether or not birds performed the display. One study briefly described in the text that the typical duration of a CSD bout is 2 seconds (Dunning et al, 2014). In our study, because there were individual variations in CSD bout duration, and not all subjects displayed CSDs to unfamiliar songs, we primarily chose the occurrence of CSDs as a measure. We then added the duration analysis in the first revision. We modified the manuscript to specify the reasons for our chosen analyses in the Methods section (pp. 12-14, lines 211-218, 222-223, 224-231).

To clarify, we calculated the total CSD duration per trial as a measure of response intensity within a trial (one song presentation), and the value was averaged across all trials in which the bird showed CSDs to a given stimulus. Thus, the mean duration reported in the text and Fig 2 is not the cumulative duration throughout the test series (we assume this is what you meant in your comment). Considering that the mean duration of CSDs was longer for the father’s song than for unfamiliar song(s) in 6 out of 7 birds, we speculate that the response frequency and response intensity measured by duration are consistent with one another. In other words, even though the difference in CSD duration between stimuli did not reach statistical significance, there was still a trend in which CSD bouts were longer for the father’s song. We hope that the revised text (p. 13-14, lines 224-231; p. 17 lines 285-289 & 295; pp. 23-24, lines 398-400) now addresses your concerns. 

3) Glmm analyses: song presentation order within a test is now a fixed factor and the repeated testing days/events (test number) a random factor. This means that you are asking the model i) to check whether each position within the test order is different from any of the other (ignoring that there is an order/time effect from the 1st to 5th position) and ii) that your analyses do not address the question of whether order and stimulus type might show an interaction. Likewise, ‘test’ is now a random factor rather than a covariate meaning that also for this variable you again omit testing of whether order or and temporal effects occur within a test series. In short, neither within a test nor in the test series, is order or time controlled statistically. The current analyses – by treating test as a random variable – will not detect an order or time effect. The reason you give in the reply for not checking for an order/time effect is saying you could not predict the direction of the effect but this argument is irrelevant- you want to test whether repeated testing leads to an effect over time (should this be the case; data inspection can show you whether there was an in- or decrease).

> Thank you for your comments. We noticed that there were still several points to be clarified in the manuscript regarding the details of the statistical analyses. Here, we give an explanation for each issue you raised:

1. The presentation order (1-5) within a test is not controlled: Since the presentation order was treated as a numerical variable, but not as a categorical factor, order and time information is retained. In other words, the GLMM has the potential to describe whether the birds’ responses could increase or decrease in earlier vs. later trials. We further clarified this type of variable (numerical) in the methods section (p. 15, lines 257-258).

2. The model does not test the interaction of stimulus type and presentation order: In the first round of the review, we explained why we decided not to include the interaction term. Please refer to our response on page 11 of the previous letter. In short, inspection of the data suggested an interaction such that the effect of presentation order was more significant in trials of unfamiliar songs. When we tried another model which included the interaction term, this resulted in a singular fit. However, even though we chose these model parameters due to a technical issue, the interaction is partly dealt with through another analysis of stimulus similarity (Fig. 3 & Table 2). Here, a GLMM was constructed to describe the factors that possibly predict the birds’ response to unfamiliar songs, and the presentation order was included as a fixed effect (again, the information of order was retained). The estimated parameters indicated a significant effect of presentation order on the occurrence of CSDs, which means that the birds showed more displays to unfamiliar stimuli presented earlier. Thus, our analyses as a whole, give substantial information about the possible interaction of stimulus type and presentation order, as far as our sample size allows.

3. The test number should be treated as a fixed effect (but not as a random effect): In the previous round of the review, we answered that it was difficult to simply assume an increase or decrease in response during the test series. Here we show the number of CSD occurrences of each bird throughout the experiment. There was no systematic change in the response frequency across individuals, suggesting that testing the effect of test number would not provide any additional information. In addition, from a technical/practical point of view, considering the overall low response rate and the individual variation in the number of tests required to finish the experiment, using a complex model with a relatively small number of subjects would likely result into an overfitting or a failure of parameter estimation. In sum, although the test number was not included as a fixed effect independent variable, the actual data indicates that doing so is not critical to the validity our statistical analysis.

In the graphs above, the total number of CSD occurrences in a test is plotted against the test number (from day1 to day5-9, depending on the individual). Each panel shows data from one female. We could not find a systematic increase or decrease in the frequency of displays throughout the test series.

Additional specific comments by line number

20-21 add ‘if the father’s song is the only song they heard’ or mention that you want to know whether it is possible at all? As is you ask for a function of something that might be an artefact of the procedure, the question of function/consequence should not be about this specific song, but whether any song (incl father) heard during this time leads to a preference? (on first principles: if there is preference learning and females learn during a sensitive phase, then if there is only one thing to learn they are likely to learn (cf. ducklings imprinting on footballs) and if these learned preferences normally affect mating preferences this is likely also the case to involve the father’s song?) Here you ask if such preference lead to matings but this has already been shown in Bengalese finches – so you should be specific here saying it is to test whether this would also happen with the father’s song?

> To ensure that our description of the research purpose is accurate, we changed the sentence as follows: ‘In the current study, we aimed to test whether the preference for the father’s song as reported in previous Bengalese finch studies, can be interpreted as a mating preference’ (p. 2, lines 20-24). Please also refer to our response to your major comment #1 for clarification of the research purpose. In addition, in Bengalese finches (as well as in zebra finches), the effect of early-life song experience on song preference at the species level has already been demonstrated by CSD assays (Clayton & Pröve, 1989), but no studies have shown that females prefer the actual (foster) father’s song to other conspecific songs. We realized that the contextualization was inadequate in the previous manuscript and revised the description in the abstract (p. 2, lines 20-24), and Introduction section (p. 4, lines 58-65 & p. 5, lines 74-77)

27 this conclusion is too strong. Again, be clear about the reductionist environment: to be sure that this statement is correct you first would have to test if this effect also occurs if the father isn’t the only tutor and if females are raised in a more natural/more socially varied environment (as you suggest in the sentences after). Adjust the statement here so it is not general but that this preference is seen in the current circumstances and make clear that to test if a learned preference for the father’s song is relevant for mate choice it should also been seen if other songs were also heard, or should likewise be seen if the females had been exposed to another song (e.g. a foster father, cage mates during the sensitive phase etc..).

> Thank you for your comment. We revised the description in the abstract, taking into consideration the specific rearing environment of the current study (p. 2-3, lines 30-34).

37 odd sentence, perhaps rephrase? (Species recognition is not something uniquely special to female songbirds but occurs across species – it is the very essence of a mating signal?)

> This sentence simply means that female birds recognize species by song. We do not argue that the conspecific-selective response is unique to females (or songbirds for that matter). To disambiguate that point, we rephrased the sentence to mention that courtship signals in general are designed to provide information on the signaler’s species (p. 3, lines 44-45).

38 independently

> Thank you for the correction. We revised this (p. 3, line 48).

39 These references do not support the statement. The sentence makes a statement about songbirds in general and that some species develop experience independent preferences but the quotes are two experimental studies in the zebra finch (the two references thus refer to only one species and this is a species where preference is NOT experience-independent, but learned to a large degree).

> We realized that the description and references here were not concordant and changed the sentence (p. 3, lines 46-47). For the latter indication, we discuss the selectivity for species-specific song features (but not about all the aspects of song preference), and the cited papers (Braaten & Reynolds, 1999; Lauay et al., 2004) both demonstrated the preference for zebra finch song vs. heterospecific song in song-naïve females. We modified the sentence so that the point is correctly conveyed.

47 or another male’s/males’ song? (can be new/different adult male or from peer group)

> We changed the description so that the father’s song is just one possible song that females may learn to prefer after early exposure (p. 4, line 57).

48ff please check the literature again: there is plenty of evidence for several species (and in particular the zf which you discuss here) that these learned preferences affect mate choice/differential allocation etc.. to say here it has not been tested ignores a lot of previous work (zebra finch, cowbirds, white crowned sparrow, song sparrow, etc.. etc..)

> Thank you. We believe our phrasing was not precise enough to sufficiently explain our logic. The issue we would like to address here is that when preference is measured by behaviors that are also expressed outside of a reproductive context, this does not necessarily mean that such preference is relevant to mating (e.g., it is possibly a more general preference for familiar stimuli). Thus, please understand that we do not argue that there is a lack of knowledge as to whether learned preferences in general affect mate choice. Instead, we argue that the choice of measurement affects what we can infer about preference. We hope that the revision here (p. 4, lines 58-65), together with related sentences in the abstract (p. 2, lines 20-22), better describe the position of this study compared to previous studies.

50 again you are only citing two zebra finch studies while making general statements about song birds – what about e.g., Darwin finches as example for pref. for father’s song?

 > Thank you for your suggestion. We added a reference on sexual imprinting in Darwin finches (Grant & Grant, PNAS, 2010 on p. 4, line 61).

56-58 you can’t address this here – drop?

> As you say, in this study we cannot address the fitness benefit of preferring the father’s song. This sentence was written, not to deal with this question, but to justify the significance of carrying out this study. We rephrased this and the following sentences to correctly convey our intention in this context (p. 5, lines 68-72). (To provide details here, we thought that it seems maladaptive for females to actually learn to prefer their father’s song as a mating signal. Choosing a male who sings exactly the same song as the female’s father may lead to inbreeding, which can greatly affect the health of the offspring. Thus, even though previous studies have shown with some types of song stimuli that the song preferences measured by general behaviors such as phonotaxis or operant responses are relevant to mate choice, it is still unclear as to whether this also applies to the father’s song. We will not know the answer until preference is measured by behaviors that are expressed specifically in the context of reproduction (i.e., CSDs).)

302 this is misleading – it should be preference for the song(s) they heard early in life – we do not know whether outside the limited exposure in the laboratory (only the father is available as tutor) which model (or several models) are influencing preferences

> We completely agree with your comment. The sentence is now revised (p. 20, line 336).

303 sweeping statement (and not correct) others have tested in both species whether song preferences translate to mate preferences – so please be specific! Perhaps what you want to say is that in BF that if such a preference for the father’s song exist that it will also lead to more CSD? It has definitely been shown for both species that song preferences (measured with a variety of methods) predict live male preferences/mating/pair formation.

> We admit that the description needs to be more specific and changed the sentences (pp. 20-21, lines 336-346). Please also refer to our response to your comment on lines 56-58 for the reason we revised the sentences this way.

316 Nicky Clayton’s work on cross fostering T.g.g and T.g.c. has shown that the females imprint on the father’s subspecies song and choose mates accordingly (the series of experiments is reviewed in (Clayton 1990).

Clayton NS 1990: Assortative mating in zebra finch subspecies, Taeniopygia guttata guttata and T. g. castanotis. Philosophical Transactions of the Royal Society of London Series B-Biological Sciences 330: 351-370. 10.1098/rstb.1990.0205

> Thank you for the comment. Although sexual imprinting at the (sub)species level has already been shown in zebra finches by Clayton (1990), it is not clear if females acquire a sexual preference for the very song (among other conspecific songs) they heard. We should have been more specific about this and thus modified the sentence accordingly (pp. 21-22, lines 360-363).

---

## [Decision Letter · Decision Letter 2]

31 Jan 2022

PONE-D-21-19382R2Female Bengalese finches show selective sexual displays to their father’s songPLOS ONE

Dear Dr. Fujii,

Thank you for submitting your manuscript to PLOS ONE. After careful consideration, we would like to invite you to submit a revised version of the manuscript that addresses the points raised during the review process.

In particular, modifications to the title and abstract (see below) are required for the manuscript to be accepted.

We look forward to receiving your revised manuscript.

Kind regards,

Jon T Sakata, PhD

Academic Editor

PLOS ONE

Journal Requirements:

Additional Editor Comments:

I thank the authors for all of their time and effort revising their manuscript. It is clear how the current research complements existing research on developmental influences on social behavior. You have seemed to satisfy most of the reviewers' concerns but before this can be accepted for publication, some modifications are required.

TITLE: The reviewer reiterates a concern about the implications of the title; i.e., that some may misinterpret the scope of the study when reading the title. While I think the title previously proposed by the reviewer might be too limited in scope, I agree that even the revised title is too broad. In my opinion, including modifiers such as “can” and descriptors such as “experimental” would appease the reviewer and my concern about the scope of the title and would communicate the major thrust of paper. I propose some alternative titles that I believe satisfy everyone’s concerns and intentions:

o “Preferences for their father’s song can manifest themselves as sexual preferences in female Bengalese finches”

o “Auditory and sexual preferences for a father’s song can co-emerge in female Bengalese finches”

o “Experimental co-development (or co-emergence) of auditory and sexual preferences for a father’s song in female Bengalese finches”

ABSTRACT: Related to the above point, numerous readers might only read the abstract. Therefore, it is important to clearly emphasize the conditions of this experiment and the scope of interpretations in the Abstract. Although there is a conclusion about the limitations of this study at the end of the abstract, the design should be made explicit early in the abstract. Therefore, the sentence should be changed in the following way (my edits in CAPS): “For this purpose, the subjects were RAISED EXCLUSIVELY with their family (I.E., DID NOT HEAR THE SONGS OF OTHER MALES) until they became sexually mature and THEN TESTED AS ADULTS.” With regard to word limits, I am confident that other parts of the abstract can be streamlined (e.g., last two sentences) to accommodate the important clarifications about the experimental design in the sentence discussed above.

o NOTE: you need to specify the meaning of the acronym CSD (on line 25)

• Include the citations about zebra finch phonotaxis and preferences as stated by the reviewer.

• Be sure to add details about the upbringing when describing the results (e.g., when ‘…attracted to their father’s song ‘add ‘if it was the only song tutor’?)

• I agree with the reviewer that the figure on page 5 of the response to reviewers would be useful to include as a figure in the main text or as a supplementary figure.

• I strongly suggest making other the editorial suggestions made by the reviewer.

Reviewers' comments:

Reviewer's Responses to Questions

**Comments to the Author**

1. If the authors have adequately addressed your comments raised in a previous round of review and you feel that this manuscript is now acceptable for publication, you may indicate that here to bypass the “Comments to the Author” section, enter your conflict of interest statement in the “Confidential to Editor” section, and submit your "Accept" recommendation.

Reviewer #2: (No Response)

2. Is the manuscript technically sound, and do the data support the conclusions?

Reviewer #2: Partly

3. Has the statistical analysis been performed appropriately and rigorously? 

Reviewer #2: (No Response)

4. Have the authors made all data underlying the findings in their manuscript fully available?

Reviewer #2: Yes

5. Is the manuscript presented in an intelligible fashion and written in standard English?

Reviewer #2: Yes

6. Review Comments to the Author

Reviewer #2: The authors wrote a detailed reply and made an effort to accommodate the comments.

Please find below a few last suggestions you might find hopefully useful.

Title: I understand your rationale for your tests and the abstract and introduction make this much clearer now, but the new title could still prime readers in the wrong way. The new title reads as if you found and describe a general property of BF mating preferences (which may or may not be the case as you had a very limited social setting in the experiment) – while a title that would acknowledge the limited song exposure they had (e.g., “female BF finches reared with father as sole song tutor show…” or similar would be a better summary of the study).

67/8 in zebra finches several studies have shown that song preferences in phonotaxis or operant tests translate to preferences in real males and that the preferences in such association tests predict pair formation (Clayton, Witte work for example).

129 here or discussion: hormone implants increase motivation but might also change preferences see for z.f.

318 ‘recognize’ doesn’t seem the right word here – ‘perceive’?

337-338 change link bewteen sentencese: the sentence in 338 starts with ‘these’ which refers to (several types of) studies in the sentence before but sentence starting in 338 is only about father’s song, so ‘these’ is not correct here.

340 ‘Other studies’ perhaps better ‘In several other species..’

342 this is one zf study and one swamp sparrow study but from context it is not clear whether you are talking about BF or other species here

347 ‘…attracted to their father’s song ‘add ‘if it was the only song tutor’?

356 is it the lab setting or the single tutor? (Situation in the lab could be similar to the wild if there were many tutors? if there is a study in the lab that can emulate social situation in the wild sufficiently you might have the same song learning although it is in the lab)

361 especially in the zf there are examples not only for species preferences but also specific songs…

The figure on page 5 in the reply is very interesting, as a reader I would appreciate seeing it either in the manuscript or in an appendix

Ref list

- species names in italics

- caps in Bengalese not used consistently

References

Acoustic characteristics, early experience, and endocrine status interact to modulate female zebra finches' behavioural responses to songs. A. Vyas, C. Harding, L. Borg and D. Bogdan

Hormones and Behavior 2009 Vol. 55 Issue 1 Pages 50-59

7. PLOS authors have the option to publish the peer review history of their article (what does this mean?). If published, this will include your full peer review and any attached files.

Reviewer #2: No

---

## [Author Response · Author response to Decision Letter 2]

11 Feb 2022

General comments:

 Thank you for your suggestions to improve the manuscript. We hope that the revised title, abstract, and the main text now address your concerns. Please see below for our specific responses. Line numbers refer to the marked-up version of the revised manuscript.

> We corrected the reference list based on the comment from the reviewer #2 and changed 2 references which give links to the figshare repository, since the article title has been modified (p. 29, refs 27 & 31).

Additional Editor Comments:

TITLE: The reviewer reiterates a concern about the implications of the title; i.e., that some may misinterpret the scope of the study when reading the title. While I think the title previously proposed by the reviewer might be too limited in scope, I agree that even the revised title is too broad. In my opinion, including modifiers such as “can” and descriptors such as “experimental” would appease the reviewer and my concern about the scope of the title and would communicate the major thrust of paper. I propose some alternative titles that I believe satisfy everyone’s concerns and intentions:

o “Preferences for their father’s song can manifest themselves as sexual preferences in female Bengalese finches”

o “Auditory and sexual preferences for a father’s song can co-emerge in female Bengalese finches”

o “Experimental co-development (or co-emergence) of auditory and sexual preferences for a father’s song in female Bengalese finches”

ABSTRACT: Related to the above point, numerous readers might only read the abstract. Therefore, it is important to clearly emphasize the conditions of this experiment and the scope of interpretations in the Abstract. Although there is a conclusion about the limitations of this study at the end of the abstract, the design should be made explicit early in the abstract. Therefore, the sentence should be changed in the following way (my edits in CAPS): “For this purpose, the subjects were RAISED EXCLUSIVELY with their family (I.E., DID NOT HEAR THE SONGS OF OTHER MALES) until they became sexually mature and THEN TESTED AS ADULTS.” With regard to word limits, I am confident that other parts of the abstract can be streamlined (e.g., last two sentences) to accommodate the important clarifications about the experimental design in the sentence discussed above.

o NOTE: you need to specify the meaning of the acronym CSD (on line 25)

> Thank you for the concrete suggestions. We modified the title and abstract so that it does not mislead the readers. We followed most of your suggestions to revise the abstract (p. 2, lines 22, 23, 27, & 29), but omitted the part “(I.E., DID NOT HEAR THE SONGS OF OTHER MALES)”, since the subject females could hear other males housed in separate cages placed in the same room (but certainly could not see nor interact with them).

• Include the citations about zebra finch phonotaxis and preferences as stated by the reviewer.

• Be sure to add details about the upbringing when describing the results (e.g., when ‘…attracted to their father’s song ‘add ‘if it was the only song tutor’?)

• I agree with the reviewer that the figure on page 5 of the response to reviewers would be useful to include as a figure in the main text or as a supplementary figure.

• I strongly suggest making other the editorial suggestions made by the reviewer.

> We changed the manuscript incorporating all these suggestions from the reviewer #2. Please see the following responses for details.

Reviewer #2:

The authors wrote a detailed reply and made an effort to accommodate the comments.

Please find below a few last suggestions you might find hopefully useful.

Title: I understand your rationale for your tests and the abstract and introduction make this much clearer now, but the new title could still prime readers in the wrong way. The new title reads as if you found and describe a general property of BF mating preferences (which may or may not be the case as you had a very limited social setting in the experiment) – while a title that would acknowledge the limited song exposure they had (e.g., “female BF finches reared with father as sole song tutor show…” or similar would be a better summary of the study).

> We understood your concern about the generality of the current results. We changed the title to “Auditory and sexual preferences for a father’s song can co-emerge in female Bengalese finches” as suggested by the Academic Editor. The short title is also revised (new title: “Auditory and sexual preferences for a father’s song in female Bengalese finches”).

67/8 in zebra finches several studies have shown that song preferences in phonotaxis or operant tests translate to preferences in real males and that the preferences in such association tests predict pair formation (Clayton, Witte work for example).

> We now added the reference ([19] Witte, Ethol Ecol & Evol, 2006) in the Introduction (p. 4, line 64) and Discussion (p. 20, line 334).

129 here or discussion: hormone implants increase motivation but might also change preferences see for z.f.

 > We additionally discussed about the possible effect of hormone implants, referring to Vyas et al., Horm Behav, 2009 (p. 21, lines 347-350), and slightly modified another sentence accordingly (p. 20, lines 339-340).

318 ‘recognize’ doesn’t seem the right word here – ‘perceive’?

> We replaced the term ‘recognize’ with ‘perceive’ as you suggested and modified the sentence according to your comment to line 347 and the editor’s advice (please see below; p. 18, lines 310-311).

337-338 change link bewteen sentencese: the sentence in 338 starts with ‘these’ which refers to (several types of) studies in the sentence before but sentence starting in 338 is only about father’s song, so ‘these’ is not correct here.

340 ‘Other studies’ perhaps better ‘In several other species..’

342 this is one zf study and one swamp sparrow study but from context it is not clear whether you are talking about BF or other species here

> To collectively address these problems, we revised the sentences (p. 20, lines 330 & 332). As the original description “their father’s song” (the boldfaced part in the sentence “These studies of female preference specifically for their father’s song measured…”) was wrong, we rather changed this part instead of the link between sentences. We also added the reference here ([19] Witte, Ethol Ecol & Evol, 2006) that you indicated in the comment to the Introduction section (line 334).

347 ‘…attracted to their father’s song ‘add ‘if it was the only song tutor’?

> We inserted the rearing condition here (p. 20, line 339) and made a similar change in a following sentence (line 341).

356 is it the lab setting or the single tutor? (Situation in the lab could be similar to the wild if there were many tutors? if there is a study in the lab that can emulate social situation in the wild sufficiently you might have the same song learning although it is in the lab)

> We acknowledge that the lab/wild distinction might not be appropriate here, and rephrased it as follows: ‘…might be captured in this unique experimental setting, but not necessarily in other social environment as discussed later in this section.’ (p. 21, lines 351-353)

361 especially in the zf there are examples not only for species preferences but also specific songs…

> We inserted a phrase here to clarify what we meant by the previous sentence (the sentence begins with ‘Although females preferred songs of the same species as their foster father, …’) (p. 21, lines 359-360)

The figure on page 5 in the reply is very interesting, as a reader I would appreciate seeing it either in the manuscript or in an appendix

> We included this figure as a new supplementary file (S3 File; p. 33, lines 561-562) and referred to it in the Method section (p. 11, lines 191-192).

Ref list

- species names in italics

- caps in Bengalese not used consistently

> We reviewed and corrected the spelling on the species names (refs 3, 17, 36, 47, 51, & 55).

---

## [Editor Report · Decision Letter 3]

17 Feb 2022

Auditory and sexual preferences for a father’s song can co-emerge in female Bengalese finches

PONE-D-21-19382R3

Dear Dr. Fujii,

We’re pleased to inform you that your manuscript has been judged scientifically suitable for publication and will be formally accepted for publication once it meets all outstanding technical requirements.

Kind regards,

Jon T Sakata, PhD

Academic Editor

PLOS ONE
---

## [Editor Report · Acceptance letter]

2 Mar 2022

PONE-D-21-19382R3 

Auditory and sexual preferences for a father’s song can co-emerge in female Bengalese finches 

Dear Dr. Fujii:

I'm pleased to inform you that your manuscript has been deemed suitable for publication in PLOS ONE. Congratulations! Your manuscript is now with our production department. 

Kind regards, 

on behalf of

Dr. Jon T Sakata 

Academic Editor

PLOS ONE